# Decoding protein methylation function with thermal stability analysis

Cristina Sayago[1], Jana Sánchez-Wandelmer[1], Fernando García [1], Begoña Hurtado [2,3], Vanesa Lafarga[4], Patricia Prieto[5], Eduardo Zarzuela[1], Pilar Ximénez-Embún [1], Sagrario Ortega[5], Diego Megías[6], Oscar Fernández-Capetillo[4], Marcos Malumbres [2,3,7] & Javier Munoz [1,8,9] ✉

Protein methylation is an important modification beyond epigenetics. However, systems analyses of protein methylation lag behind compared to other modifications. Recently, thermal stability analyses have been developed which provide a proxy of a protein functional status. Here, we show that molecular and functional events closely linked to protein methylation can be revealed by the analysis of thermal stability. Using mouse embryonic stem cells as a model, we show that Prmt5 regulates mRNA binding proteins that are enriched in intrinsically disordered regions and involved in liquid-liquid phase separation mechanisms, including the formation of stress granules. Moreover, we reveal a non-canonical function of Ezh2 in mitotic chromosomes and the perichromosomal layer, and identify Mki67 as a putative Ezh2 substrate. Our approach provides an opportunity to systematically explore protein methylation function and represents a rich resource for understanding its role in pluripotency.

Among other features, pluripotent ESCs are characterized by a derestricted epigenetic state contributing to the onset of developmental programs. For instance, mESCs possess bivalent histone marks, defined by the co-occurrence of activating H3K4me3 and repressive H3K27me3, at the promoters of developmental genes[1]. However, recent proteomic analyses in cancer cells have firmly established that protein methylation fine-tunes numerous important protein functions beyond histones[2], such as DNA replication, protein synthesis, RNA metabolism and signal transduction[3,4]. Not surprisingly, protein methylation is also becoming recognized as an important regulatory mechanism in pluripotency. For instance, the key pluripotency protein Transcription factor Sox2 (Sox2) is methylated by Prmt4 (Carm1), promoting its association with chromatin[5]. Sox2 protein levels in ESCs are also regulated by a balanced methylation and phosphorylation

switch[6]. The Histone methyl-transferase Setd7 methylates Sox2, which is subsequently ubiquitinated and degraded by the proteasome, thereby inhibiting Sox2 transcriptional activity. This effect is counterbalanced by the phosphorylation of Sox2 by the protein kinase Akt1, preventing its methylation and degradation. The importance of methylation in pluripotency is also underscored by tight regulation of S-adenosylmethionine (SAM) levels in ESCs[7,8]. All these data suggest that ESCs might possess a singular regulation of protein methylation. However, a proteome-wide characterization of this modification in pluripotent cells is lacking.

Typically, protein methylation is studied by directly quantifying methylated peptides in response to cellular perturbation. Although this approach has provided valuable information on protein methylation networks[4,9,10] it presents some caveats because it implies tedious

[1]Proteomics Unit, Spanish National Cancer Research Centre (CNIO), 28029 Madrid, Spain. [2]Cell Division and Cancer Group, Spanish National Cancer Research Centre (CNIO), 28029 Madrid, Spain. [3]Cancer Cell Cycle group, Vall d'Hebron Institute of Oncology (VHIO), 08035 Barcelona, Spain. [4]Genomic Instability Group, Spanish National Cancer Research Centre (CNIO), 28029 Madrid, Spain. [5]Mouse Genome Editing Unit, Spanish National Cancer Research Centre (CNIO), 28029 Madrid, Spain. [6]Confocal Microscopy Unit, Spanish National Cancer Research Centre (CNIO), 28029 Madrid, Spain. [7]Catalan Institution for Research and Advanced Studies (ICREA), 08010 Barcelona, Spain. [8]Cell Signaling and Clinical Proteomics Group, Biocruces Bizkaia Health Research Institute, 48903 Barakaldo, Spain. [9]Ikerbasque, Basque foundation for science, 48011 Bilbao, Spain. ✉e-mail: javier.munozperalta@osakidetza.eus

and cost-effective immune-purification procedures and the difficulties in identifying comprehensively methylated peptides by MS[11,12]. Moreover, multiple methylation sites often co-exist in the same peptide, making the assignment of site-specific changes rather complex[9]. Recently, new methodologies have been developed to detect changes in the thermal stability of proteins[13]. Mounting evidence shows that alterations in this biophysical parameter can reveal important molecular events such as post-translational modifications (PTMs), protein-protein interactions and other types of structural re-arrangements[14]. Consequently, these approaches are being exploited to study diverse biological processes such as cell cycle[15], protein aggregation[16] and even inferring protein functions[17]. Here, we sought to extend this approach to study protein methylation. Using mESCs as a cell model system, we have implemented an approach based on thermal protein stability and demonstrate that this biophysical parameter serves as a proxy to identify potential proteins and functions controlled by methylation.

## Results

### The non-histone methyl proteome landscapes of mESCs and MEFs

To understand the scope of functions and processes regulated by protein methylation in mESCs, we first sought to define a protein expression map of methyl-transferases and de-methylases in these cells and compared that to mouse embryonic fibroblasts (MEFs). We obtained nearly complete proteomes for both mESCs and MEFs (Supplementary Data 1) and established unambiguous identification for 92% of the methyl-transferases and de-methylases encoded in the genome (Fig. 1A and Supplementary Fig. 1A), including several protein isoforms[18] (Supplementary Data 2). This analysis revealed that most of these enzymes were higher expressed in mESCs than in MEFs which we validated in additional mESCs cell lines and differentiated tissues by PRM (Supplementary Fig. 1B and Supplementary Data 3). Next, we aimed to catalogue downstream methylated proteins in mESCs and

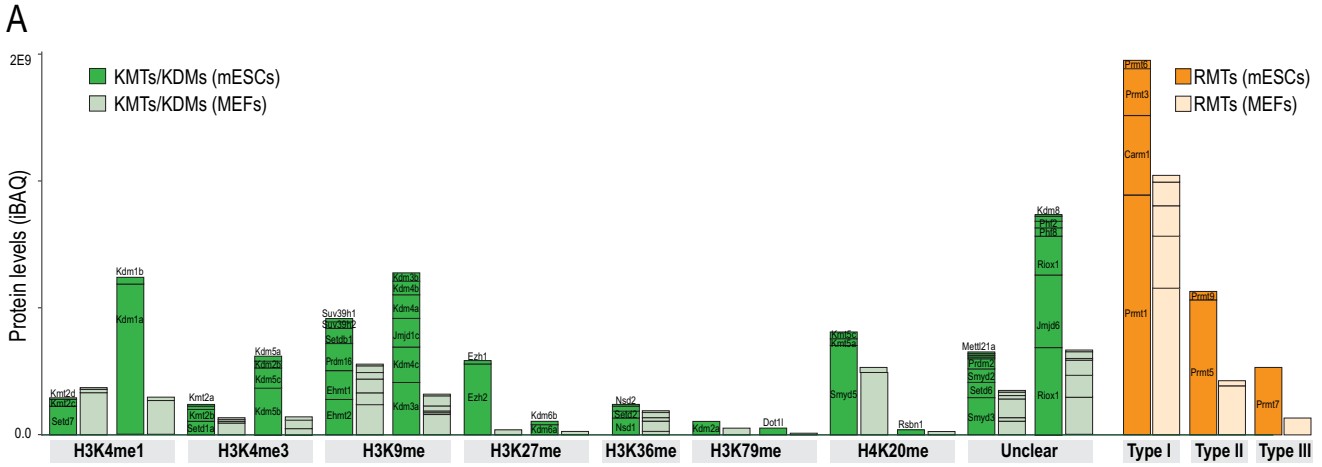

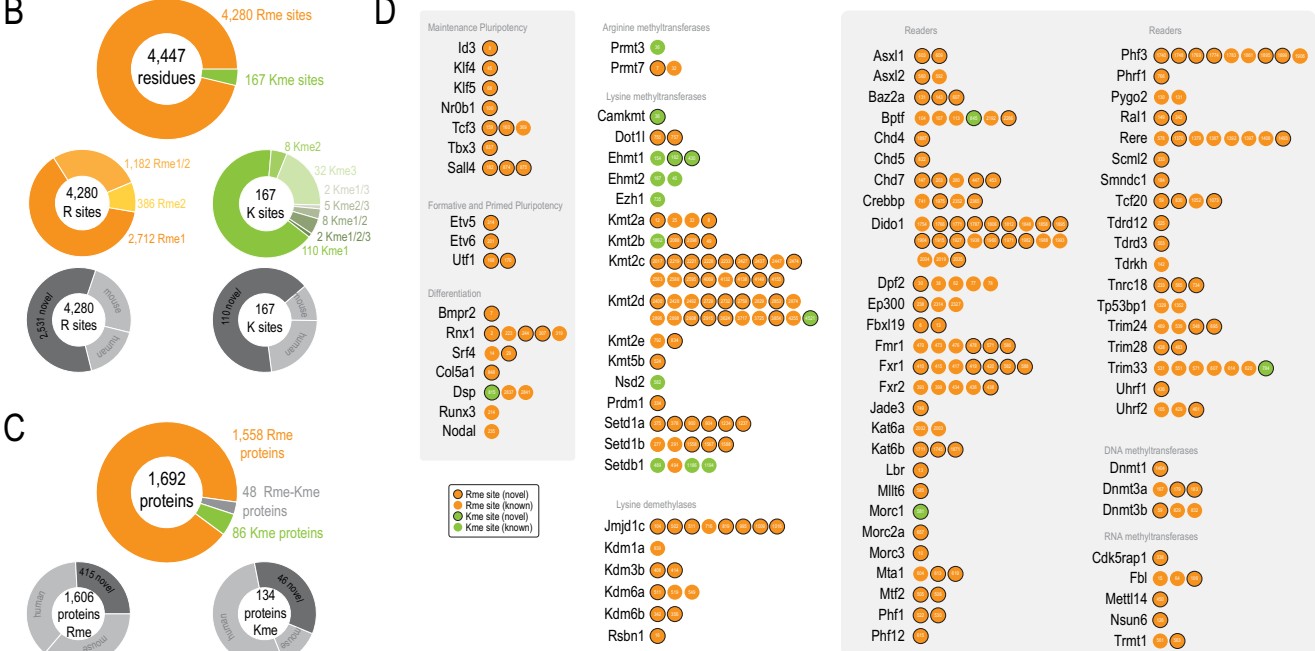

**Fig. 1 | Proteomic expression maps of methyl-transferases and de-methylases in mESCs and differentiated cells. A** estimated protein levels (iBAQ) of KMTs, KDMs and RMTs in mESCs and MEFs classified on the basis of their substrate specificity. **B, C** Number of methylated sites and proteins in their different forms identified in our data sets. Known methylated sites and known methylated proteins in mouse and human were retrieved from www.phosphosite.org. **D** examples of proteins identified as Lys or Arg methylated. The specific residues are shown inside the circles. Outlined circles are novel sites.

MEFs using pan-specific antibodies and identified 4,280 Rme and 167 Kme sites (Fig. 1B, C and Supplementary Data 4). Despite the depth of our analysis, we noticed a slight bias towards abundant proteins. Also, numerous highly-charged spectra remained unidentified because of the sup-optimal HCD fragmentation mode used here[19] (Supplementary Fig. 2), suggesting that the methyl-proteome is still under-sampled by current MS-based approaches. Nevertheless, 59–65% of the sites and 25–34% of the proteins were novel, indicating that the catalogue of Rme and Kme is far from complete. GO analyses showed that Arg methylation is most prominently involved in RNA-related processes (Supplementary Data 5). In addition, we found an enrichment in development and cell differentiation terms, including methylations in several factors involved in pluripotency (Fig. 1D). Both Arg and Lys methylated proteins were enriched in chromatin remodeling and histone modification, including numerous methylations in RMTs, KMTs and KDMs as well as methyl readers and DNA/RNA methyl-transferases (Fig. 1D) supporting the presence of auto-regulatory effects. As expected, Arg sites were particularly enriched in Arg and Gly (Supplementary Fig. 3A). However, only 10% of Arg sites were present in canonical RG motifs[20], while 25% of sites laid in non-canonical RG motifs (Supplementary Fig. 3B). Canonical-RG motifs were highly enriched in disordered regions of proteins in agreement with previous reports[19] (Supplementary Fig. 3C) and showed higher stoichiometries (Supplementary Fig. 3D). Interestingly, 65% of the Arg sites localized in non-RG sequences, with 46% of the identified proteins containing only non-RG motifs (Supplementary Data 6), which suggests that RMTs might have mutually exclusive substrates. We compared methylation levels between mESCs and MEFs with total protein abundance and found a moderate correlation (Supplementary Fig. 4A), implying that the differences in methylation are explained, only to a certain extent, by differences in protein expression. Using a conservative 16-fold change threshold, we defined 462 sites (243 proteins) enriched in mESCs and 342 sites (225 proteins) in MEFs, with only 36 proteins in common (Supplementary Fig. 4B). However, GO analyses revealed remarkably similar functions between both cell types. These results suggest that mESCs and MEFs possess unique protein methylation signatures controlling however similar biological processes.

## Prmt5 inhibition in mESCs leads to thermal stability changes of numerous Arg-methylated proteins, including known-substrates

Having defined the landscape of enzymes and substrates involved in protein methylation in mESCs, we next aimed to deconvolute their complex functional relationships and hypothesized that changes linked to protein methylation might alter the thermal stability of proteins. To test this idea, we first measured thermal stability changes caused by Protein arginine N-methyltransferase 5 (Prmt5) inhibition in mESCs. Prmt5 is essential for pluripotency[21], and it is the major type II RMT in mESCs (Fig. 1) with numerous well-characterized substrates reported in the literature[9,10,22]. Treating mESCs with 50 nM of GSK591 and GSK595 for two days was sufficient to reduce the symmetric dimethylation of Snrpd3, a known Prmt5 substrate (Supplementary Fig. 5). To identify potential changes in the thermal stability of proteins, we implemented the Proteome Integral Stability Alteration (PISA)[23] assay using a refined range of temperatures of 51–56 °C[24] (Fig. 2A). In parallel, cells were lysed in 5% SDS to assess for potential changes in protein abundance. Overall, we determined stability and abundance changes for 8112 proteins (Supplementary Data 7). PCA showed a clear separation of GSK591 and GSK595 samples from DMSO (Supplementary Fig. 6) with both inhibitors affecting the stability of numerous proteins in a very similar manner (Fig. 2B). Using a moderated t-test, we found 530 and 548 proteins that exhibited enhanced stability in response to GSK591 and GSK595, respectively (FDR < 5%) (Supplementary Fig. 7). Importantly, 352 proteins increased stability in common to both inhibitors (p.val 8E-331; hypergeometric test),

indicating a very similar response (Supplementary Fig. 7). Conversely, 446 proteins decreased stability in response to both inhibitors (p.val 2E-389; hypergeometric test). In addition, we found that both inhibitors induced reproducible changes in protein abundance (Fig. 2B). Nevertheless, protein stability and abundance changes did not correlate (Fig. 2C), indicating that GSK591 and GSK595 alter the stability of multiple proteins independently of gene expression changes. Drug engagement often increases the thermal stability of target proteins[25]. Indeed, both Prmt5 and its canonical partner Methylosome protein 50 (Wdr77)[26] showed an increase in stability in response to both inhibitors with no apparent changes in protein levels (Fig. 2D). Remarkably, the stability of the other PRMTs was not affected, except Prmt6 (type I) whose expression is 15-fold lower than Prmt5 in mESCs (Fig. 1), suggesting that a great part of the observed changes in thermal stability in our data might be attributed to the specific inhibition of Prmt5.

Next, we investigated the impact of Prmt5 inhibition on the stability of proteins identified as Arg methylated in our immuno-purifications. We found that a significant fraction of Arg-methylated proteins showed increased thermal stability (Fig. 2E). The same analysis on the subset of ADMA and SDMA proteins also showed an increase in stability, with SDMA showing the largest effect (Fig. 2E). The presence of ADMA in 45% of the SDMA-containing proteins (Supplementary Data 4), might explain the increase in stability of ADMA. Importantly, the abundance of methylated proteins barely changed in response to both inhibitors (Fig. 2E). These results indicate that inhibition of Prmt5 alters the stability of numerous Arg methylated proteins, including SDMA, in agreement with Prmt5 being the main type II PRMTs in mESCs. Several studies have reported the identification of Prmt5 substrates using immuno-purification followed by MS[9,10]. Despite obvious differences between cell lines, our mESCs data showed changes in the stability in 8 of the 12 Prmt5 substrates reported in human AML cells[10] and 10 of the 57 substrates reported in HeLa cells[9] (Fig. 2F). Further, Arg methylation is involved in protein-protein interactions and some Prmt5 substrates, including the Sm protein B/B' (Snrpb) and the THO complex subunit 4 (Alyref), physically and stably interact with Prmt5[27]. We retrieved Prmt5 interactors from Biogrid and found 50 of them with altered stability in our data, including several epigenetic regulators and methyl-binding proteins (Fig. 2F). These results show that thermal stability analysis enables the identification of Prmt5 substrates, complementing classic immuno-purification-based approaches.

## G3bp2 is a Prmt5 substrate and is symmetrically di-methylated in its c-terminus disordered region

The Ras GTPase-activating protein-binding protein 2 (G3bp2) increased its thermal stability upon Prmt5 inhibition and was found in our immune-precipitation data to be methylated in several Arg, including SDMA located in a low complexity region enriched in RG motifs in its C-terminus (Fig. 3A). We performed an in vitro methyl-transferase assay using recombinant G3bp2 and Prmt5-Wdr77 and analysed the reaction by LC-MS/MS. This revealed the presence of mono and di-methylation in R438 and R468 of G3bp2, in great agreement with a recent report[28] (Fig. 3B and Supplementary Fig. 8A). Importantly, GSK595 prevented these methylations, and we did not identify any methylation in G3bp2 alone, confirming the assay's specificity. Moreover, the decrease of the unmodified counterpart peptide upon methylation by Prmt5 indicates that R468me is highly stoichiometric (i.e. high fractional occupancy) (Fig. 3C). On the other hand, the increase of R468me1 in the presence of GSK595 (Fig. 3B) is consistent with the distributive mode of protein substrate methylation by Prmt5[29] (Supplementary Fig. 8B). In view of these results, we next aimed to measure the actual impact of Arg methylation on the protein thermal stability using this in vitro methyl transferase assay, as this recapitulates a simplified methylated and non-methylated model system. To this end, we spiked-in methylated and unmethylated G3bp2 over a

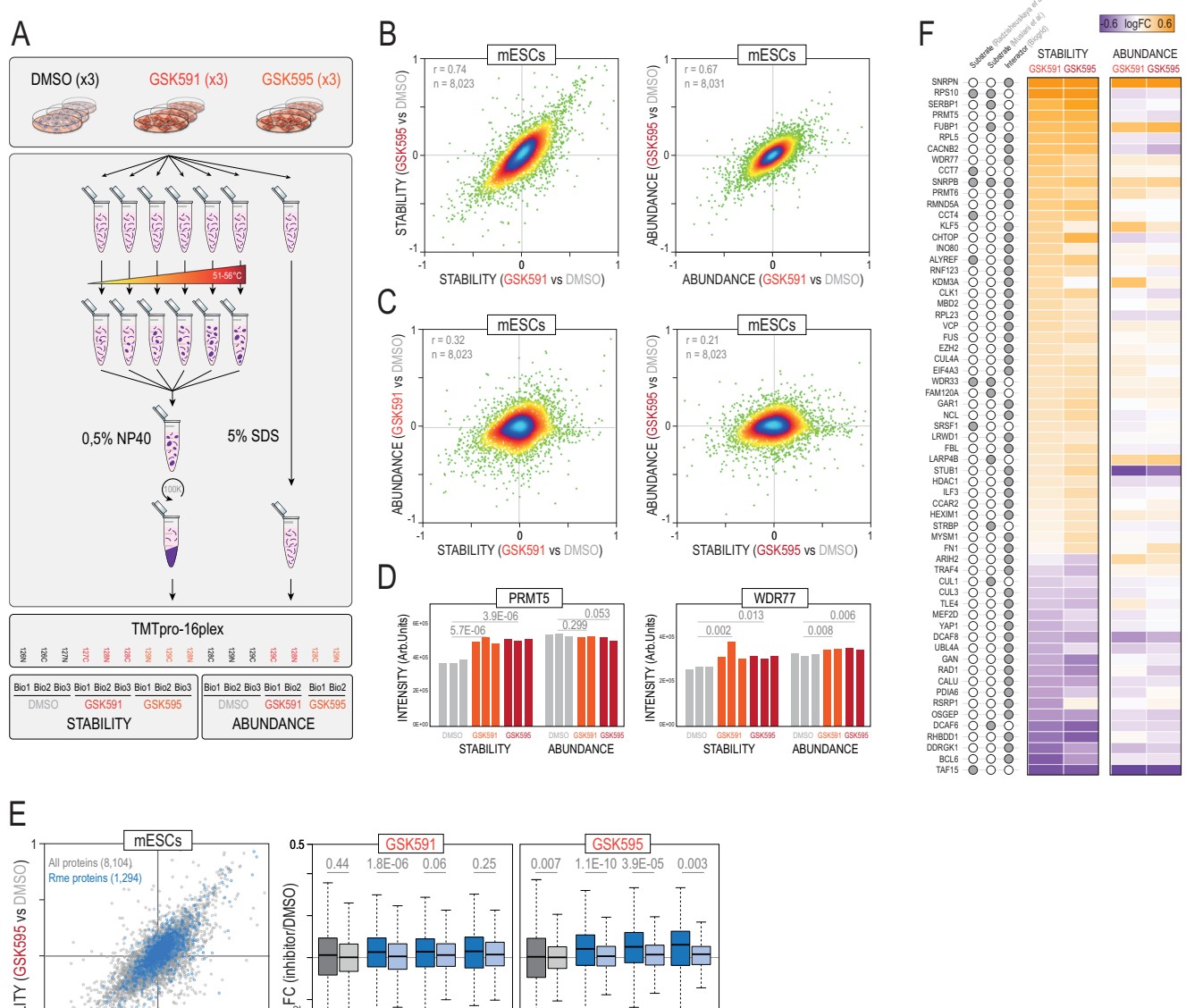

**Fig. 2 | Thermal proteome analysis of mESCs upon inhibition of Prmt5. A** mESCs were treated with two Prmt5 inhibitors (GSK591 and GSK595) or DMSO and subjected for the proteome integral solubility alteration (PISA) assay. In parallel, cells were lysed in SDS to estimate total protein abundance levels. Samples were TMTpro-labelled as shown in the scheme and analysed by LC-MS/MS. Three biological replicates were performed for all conditions except for the abundance of GSK591 and GSK595 which included two biological replicates. **B** Density scatterplots showing the correlation between GSK591 and GSK595 for stability (left) and abundance (right) changes. **C** Density scatterplots showing the correlation between stability and abundance changes for GSK591 (left) and GSK595 (right). **D** Stability and abundance levels for Prmt5 (left) and its partner Wdr77 (right). *P*-values were calculated using a moderated t-test (*limma*) (two-sided) (*n* = 3 biological

replicates). **E** Scatterplot (left) showing the stability changes for the 1294 proteins identified in this study as Arg methylated (in blue); box-plots (right) showing the distribution of stability (STA) and abundance (ABU) log₂ ratios for all quantified proteins, Arg methylated proteins (Rme), asymmetric (ADMA) and symmetric Arg (SDMA) methylated proteins. *P*-values were calculated using a non-parametric Wilcoxon test (two-sided) (stability vs abundance) and a Mann Whitney U test (two-sided) (stability vs All proteins). For box-plots, median is shown; box limits indicate the 25th and 75th percentiles; whiskers extend 1.5 times the interquartile range from the 25th and 75th percentiles. Sample size in each condition (n) is shown below in parenthesis. **F** Heatmap showing the stability and abundance changes upon inhibition of Prmt5 in mESCs for known Prmt5 substrates[9,10] and for Prmt5 interactors (thebiogrid.org). Source data are provided as a Source Data file.

background proteome (SF9 cells from *Spodoptera frugiperda*) (Fig. 3D) which acted as a carrier and was also used for normalization purposes. We subjected the protein mixture to 5 different temperatures and identified the remaining soluble protein fraction after centrifugation by LC-MS/MS using DDA. As expected, insect proteins denatured and aggregated with incremental temperatures (Fig. 3E). Both Prmt5 and Wdr77 showed similar denaturing profiles (Fig. 3F, G). Importantly,

both proteins exhibited a significant increase in thermal stability in the presence of GSK595 (particularly evident at 53 and 55 °C), confirming and validating the target engagement of this compound on the methylosome complex (Fig. 3F, G). To accurately quantify G3bp2 levels in these experiments, we used PRM. Melting curves, however, were rather similar for methylated and unmethylated G3bp2 (Fig. 3H) (see discussion).

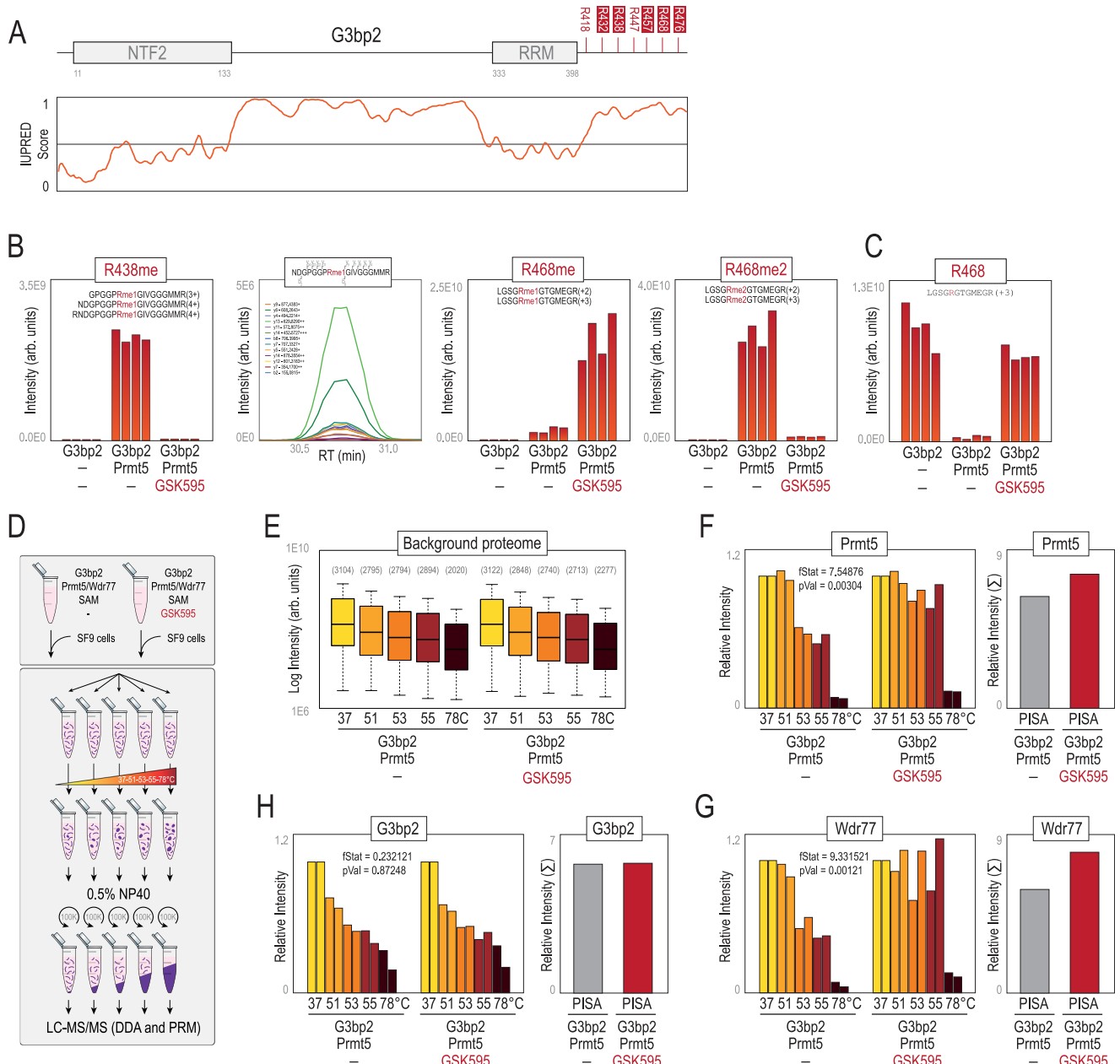

**Fig. 3 | Prmt5 dimethylates G3bp2 at R468. A** Pfam domains and prediction of low complexity regions for G3bp2. Known Rme sites (PhosphositePlus) in G3bp2 are shown: those identified in our data in mESCs are highlighted in red squares. **B** Quantification of Rme levels (R438me, R468me and R468me2) identified in G3bp2 in an in vitro methyl-transferase assay using Prmt5. One example of the extracted ion chromatogram (XIC) of a methylated peptide analysed in Skyline is shown. **C** Quantification of the unmodified counterpart peptide containing R468. **D** Experimental design to measure the impact of the methylation status of G3bp2 on its thermal stability. **E** Quantification of insect proteins (SF9 cells) across all 5 temperatures used as carrier in both experiments. Soluble Prmt5 (**F**), Wdr77 (**G**) and G3bp2 (**H**) levels detected in each temperature in both in vitro methyl-transferase assays (with and without GSK595). P-values were calculated using the nonparametric analysis of response curves (NPARC) (two-sided)[76] ($n = 2$ biological replicates). On the right, the sum of all data points is presented (similar to PISA) for each protein. For box-plots, median is shown; box limits indicate the 25th and 75th percentiles; whiskers extend 1.5 times the interquartile range from the 25th and 75th percentiles. Sample size in each category is shown below in parenthesis. Source data are provided as a Source Data file.

## Prmt5 regulates numerous proteins involved in RNA biology and promotes stress granules assembly

To gain insight into the biological processes controlled by Prmt5, we performed GO analyses. We found a clear enrichment in processes related to RNA biology among the proteins with increased stability, in agreement with the known role of Prmt5 in mRNA splicing[30,31] (Supplementary Data 8). Destabilized proteins, on the other hand, showed a lower degree of functional co-regulation including general metabolic processes and several subunits of the electron transport chain. We noticed, however, that these mitochondrial proteins also decreased in abundance indicating that these were not genuine stability alterations but rather changes in gene expression. To investigate stabilized proteins in more detail, we examined a recent list of mRNA-binding proteins identified in mESCs[32] and found that many were significantly stabilized in response to Prmt5 inhibition (Fig. 4A). Interestingly, we did not see a similar trend for proteins with an affinity toward non-polyadenylated RNA (i.e., tRNA, rRNA, snRNA and snoRNA) as determined by Trendel et al.[33], suggesting that Prmt5 mainly regulates processes linked to mRNAs. Our data also showed a high enrichment in ribosome subunits (with 67/75 identified subunits stabilized) and

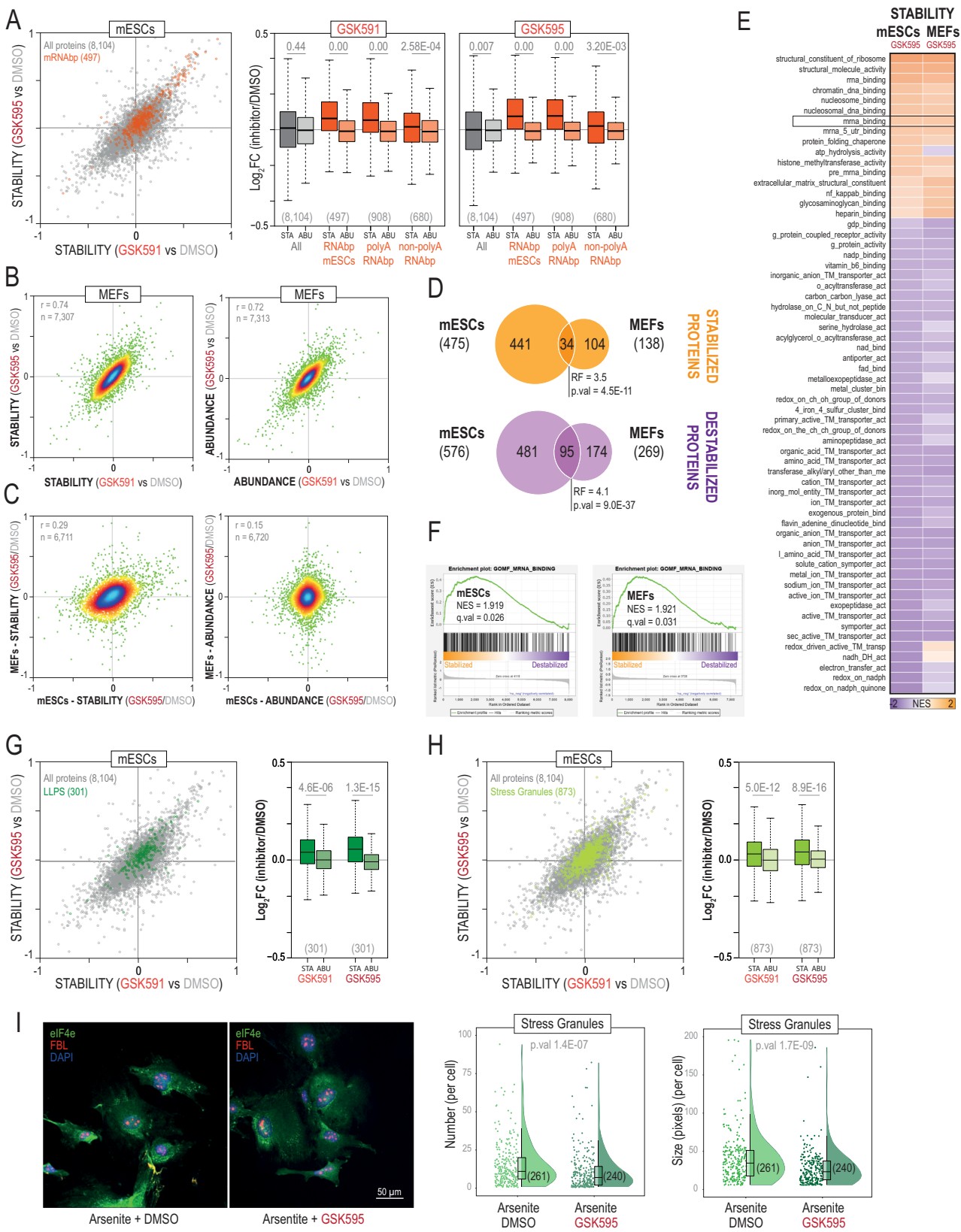

numerous ribosome biogenesis proteins, in agreement with the known roles of Prmt5 in Rps10 methylation and promoting ribosome biogenesis[34]. Importantly, our data revealed potential hits and processes less-well known for Prmt5 (Supplementary Data 8). For instance, among the stabilized proteins, we found several subunits of NuRD (Mta2, Mta3, Mbd2, Gatad2a and Hdac1) and PRC2 (Ezh2, Suz12, Eed

and Mtf2) supporting recent studies that linked Prmt5 with these repressive complexes[35,36]. Surprisingly, we also found a significant enrichment in proteins involved in protein folding. Among them were all 8 subunits of the TRiC/CCT chaperonin complex. Our data, therefore, confirm the findings by Radzisheuskaya et al., who showed that subunit CCT4 is a well-validated Prmt5 substrate[10]. Collectively, these

**Fig. 4 | Prmt5 regulates numerous mRNAbp including liquid-liquid phase separation proteins involved in stress granules formation. A** Scatterplot (left) showing the stability changes measured in mESCs in response to Prmt5 inhibition for 497 proteins reported by Kwon et al.[32] as RNAbp in mESCs identified in our study (in orange); box-plots (right) showing the distribution of stability (STA) and abundance (ABU) log$_2$ ratios for all quantified proteins, RNAbp identified by Kwon et al.[32], polyA-RNAbp and non-polyA-RNAbp identified by Trendel et al.[33]. **B** Density scatterplots showing the correlation between GSK591 and GSK595 for stability (left) and abundance (right) changes in response to Prmt5 inhibitors in MEFs. **C** Density scatterplots showing the correlation between mESCs and MEFs for stability (left) and abundance changes (right). **D** Overlap between stabilized and destabilized proteins found in mESCs and MEFs. P-values are calculated using a hypergeometric test (two-sided). RF, representation factor. **E** Heatmap showing the normalized enrichment scores (NES) for molecular functions in the thermal stability analysis.

**F** GSEA plots for proteins annotated as mRNA binding. **G, H** Scatterplot (left) showing the stability changes measured in mESCs in response to Prmt5 inhibition for 301 proteins regulated by LLPS (Uniprot) (**G**) and for 873 proteins involved in stress granules (**H**); box-plots (right) showing the distribution of stability (STA) and abundance (ABU) log$_2$ ratios for those proteins. **I** eiF4e (stress granules marker) and FBL (nucleolus marker) levels in arsenite-stressed MEFs with and without GSK595. The number of stress granules (per cell) and their sizes is shown on the right in the form of violin plots. Sample size (independent measurements) in each condition (n) is shown below in parenthesis. For box-plots, median is shown; box limits indicate the 25th and 75th percentiles; whiskers extend 1.5 times the interquartile range from the 25th and 75th percentiles. Sample size in each category is shown below in parenthesis. *P*-values are calculated using a non-parametric Wilcoxon test (two-sided) (**A, G, H**) and U Mann-Whitney U test (two-sided) (**I**). Source data are provided as a Source Data file.

results confirm known roles of Prmt5 and expand its potential functions and targets. Therefore, we conclude that thermal stability analysis can be used to identify biological information linked to Arg methylation function.

Prmt5 is essential to maintain pluripotency[21], and its protein expression in MEFs is significantly lower than mESCs (Fig. 1). To better understand the complexity of functions controlled by Prmt5, we next profiled thermal stability changes upon Prmt5 inhibition in MEFs. Using the same experimental conditions as in mESCs (Supplementary Fig. 9), we found that both compounds changed the stability of numerous proteins in MEFs and that these changes did not respond to differences in protein abundance (Fig. 4B and Supplementary Fig. 7). Moreover, we found that both Prmt5 and its partner Wdr77 were stabilized (Supplementary Data 8), and the overall stability of proteins identified as methylated in our data sets also increased significantly in MEFs (Supplementary Fig. 10). Next, we compared stability changes caused by Prmt5 inhibition in mESCs and MEFs and found a moderate correlation, whereas protein abundance showed no correlation (Fig. 4C). The number of proteins stabilized or destabilized in common to both cell types significantly overlapped (Fig. 4D). Yet, a larger fraction of proteins was exclusively affected in mESCs and MEFs, with mESCs showing a major response to Prmt5 inhibition in terms of the number and the magnitude of changes (Supplementary Fig. 7). Despite these differences, GSEA analyses showed however a general agreement in the functions affected by Prmt5 inhibition in both cell types with multiple RNA-related processes among the proteins with increased stability (Fig. 4E, F). These results indicate that Prmt5 controls similar functions related to RNA biology in mESCs and MEFs by regulating common but also different proteins.

Arg methylation regulates the formation of membrane-less organelles via liquid-liquid phase separation (LLPS) mechanisms[37]. These include stress granules, which are ribonucleoprotein assemblies formed in the cytoplasm to prevent mRNA degradation under different types of stress. In our data, inhibition of Prmt5 led to an overall increase in the stability but not the abundance of numerous proteins involved in LLPS and proteins associated with stress granules[38,39] in both mESCs and MEFs (Fig. 4G, H and Supplementary Fig. 10). These results suggest that Prmt5 might be involved in regulating of these membrane-less organelles. To test this, we treated MEFs with arsenite, a potent inducer of stress granules via oxidative stress. As expected, this led to the appearance of Eif4e-positive cytoplasmic aggregates (Fig. 4I), a well-known stress granule marker. GSK595 alone, however, did not promote the formation of stress granules but led to alterations in nucleolar fragmentation (Supplementary Fig. 11), which is consistent with the thermal stability changes found in nucleolar proteins (Supplementary Data 6). Interestingly, the addition of GSK595 to arsenite-stressed cells significantly decreased the number of Eif4e-positive foci, which were also smaller (Fig. 4I). We conclude that methylation of Prmt5 substrates promotes stress granules formation, in agreement with previous data showing that SDMA in the RGG domain of the U6

snRNA-associated Sm-like protein (Lsm4) stimulates processing body formation[40].

## Thermal stability analysis in response to Ezh2 inhibition reveals an epigenetic cross-talk between H3K27me3 and other histone marks

Although several examples demonstrate the importance of Lys methylation beyond epigenetics[41], our knowledge on this matter is rather limited. The Histone methyl-transferase Ezh2 is the catalytic subunit of the Polycomb Repressive Complex 2 (PRC2), involved in the deposition of H3K27me3[42] but also methylates non-histone proteins[43]. To gain insights into the methyl-transferase functions of Ezh2 in pluripotency, we measured stability changes upon Ezh2 inhibition in mESCs and compared them to those observed in MEFs. We found that 2 μM GSK126 and 0.5 μM EPZ-6438 for two days decreased the levels of H3K27me3 in both cell types (Supplementary Fig. 12). Under these conditions, we found that both compounds induced changes in thermal stability and abundance of numerous proteins in mESCs, with GSK126 causing almost three times more changes than EPZ-6438 (Fig. 5A and Supplementary Data 7), in agreement with GSK126 being the most potent Ezh2 inhibitor[44]. Importantly, stability changes were not determined by abundance levels (Supplementary Fig. 13). Among the proteins with increased stability in mESCs in response to GSK126, we found Ezh2 (Supplementary Fig. 14). EPZ-6438, on the other hand, did not affect Ezh2 stability. Yet, the overall stability profiles correlated (Fig. 5A), and the vast majority of GSK126 changes were reproduced by EPZ-6438 (Supplementary Fig. 13), indicating that both compounds caused a similar biological response but of a different magnitude. Remarkably, Ezh2 inhibition in MEFs caused far fewer stability changes (Supplementary Fig. 13). Of note, Ezh2 itself was undetected amongst the 8,016 proteins identified in MEFs, underscoring its role in pluripotency. In mESCs, all PRC2 subunits, with the exception of PRC2.2 subunits, showed stability changes in GSK126, with the Histone-binding protein Rbbp4 and the Polycomb protein Suz12 also changing in response to EPZ-6438 (Supplementary Fig. 14). These results indicate that GSK126 and EPZ-6438 induce an important alteration in the stability of the mESCs proteome, including the PRC2 complex.

Using GO analyses, we found that destabilized proteins showed an enrichment in nucleotide metabolism, consistent with a recent report showing that Ezh2 regulates GTP production in melanoma cells via the Inosine-5'-monophosphate dehydrogenase 2 (Impdh2)[45]. On the other hand, stabilized proteins in mESCs were enriched, among other functions, in chromatin organization (Supplementary Data 9 and Supplementary Fig. 15C). For instance, we found several KMTs, KDMs and even Lys acetyl-transferases with increased thermal stability in response to both inhibitors. Given the major decrease in H3K27me3 upon Ezh2 inhibition (Supplementary Fig. 12), we reasoned that protein readers for this epigenetic mark might exhibit altered stability in

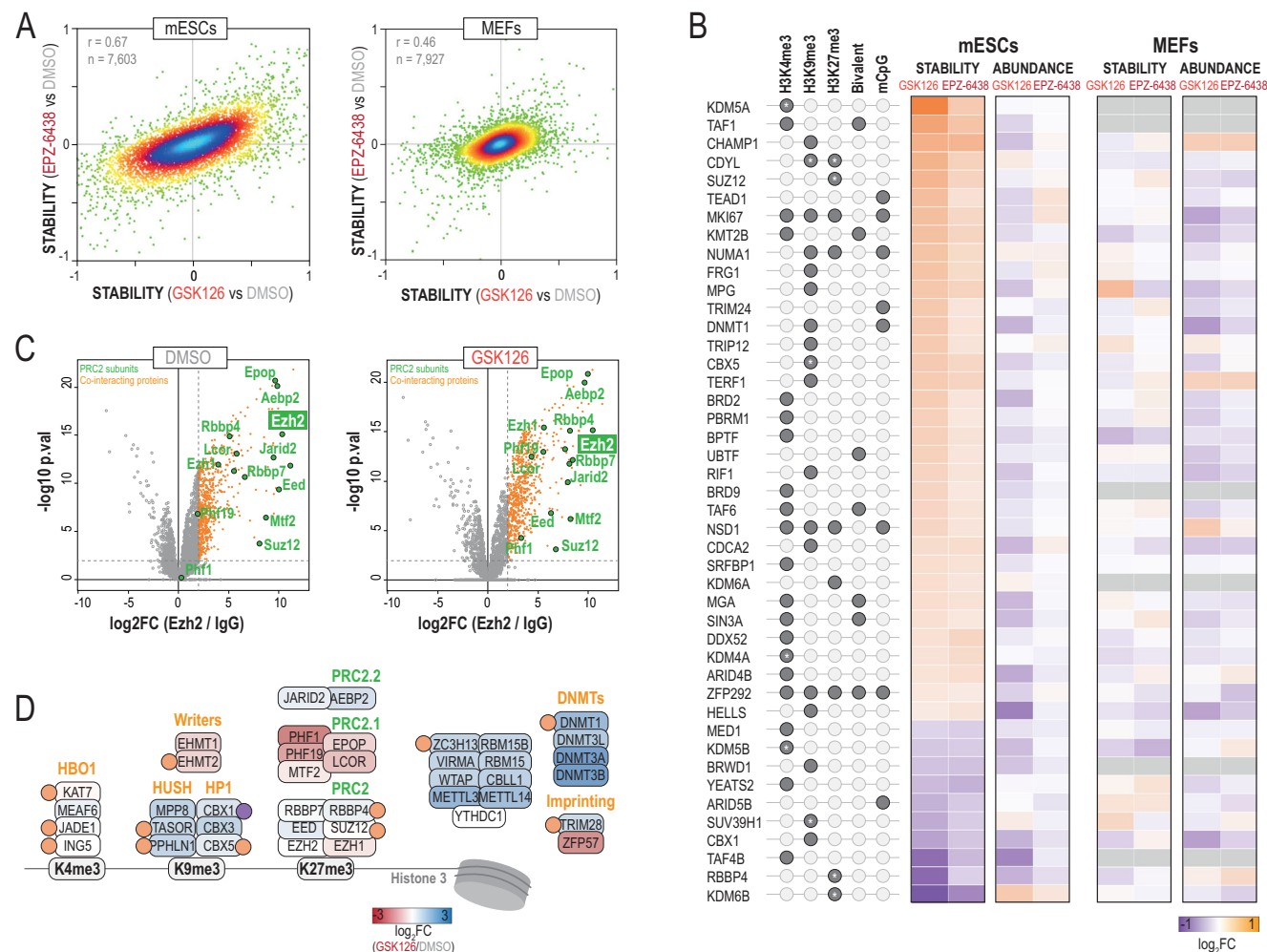

**Fig. 5 | Thermal stability changes in numerous epigenetic regulators upon Ezh2 inhibition in mESCs. A** Density scatterplots showing the correlation between stability changes in GSK126 and EPZ-6438 for mESCs (left) and MEFs (right). **B** Heatmap showing the log2 ratios for stability and abundance levels of proteins associated to different epigenetic marks in mESCs. **C** Volcano plots showing the identified Ezh2 interacting proteins (FDR < 5%) in mESCs in DMSO (left) and GSK126 (right) (n = 3 biological replicates). **D** Schematic showing some Ezh2 interacting proteins with known functions in epigenetics. The colour code represents the difference in interaction found between DMSO and GSK126. Proteins with altered thermal stability in response to Ezh2 inhibition in mESCs are also labelled with a circle (orange, increased; purple, decreased). Source data are provided as a Source Data file.

our data. Thus, we examined proteins known to associate with specific histone modifications in mESCs[46]. Among the 16 readers reported for H3K27me3[4], four of them (the Nuclear mitotic apparatus protein Numa1, the Proliferation marker Mki67, the Histone methyl-transferase Nsd1 and the Zinc finger protein Zfp292) showed increased stability in mESCs but not in MEFs (Fig. 5B). In mESCs, H3K27me3 is found with H3K4me3 at developmental gene promoters[1]. Our data revealed that seven readers of this bivalent mark also increased their thermal stability, as well as 21 proteins associated with monovalent H3K4me3 (Fig. 5B). Several proteins typically associated with H3K9me3 also bind H3K27me3[46]. Our thermal stability analysis revealed alterations in 16 proteins associated to H3K9me3, including two of the three heterochromatin proteins 1 (HP1) (the Chromobox proteins Cbx1 and Cbx5) and the Chromodomain Y-like protein Cdyl[47].

To validate the potential connections between Ezh2 and some of these epigenetic regulators, we purified endogenous Ezh2 from mESCs and identified its interacting proteins by LC-MS/MS. As expected, all the PRC2 subunits were identified (Fig. 5C) at the known stoichiometries[42] (Supplementary Fig. 16A), indicating a high specificity in our immuno-purification. Importantly, this analysis also revealed 890 potential Ezh2 co-interacting proteins (Fig. 5C and Supplementary Data 10). Among them, we found all the subunits of the

HBO1 complex, an H3K4me3 reader involved in H3 and H4 acetylation near transcription start sites[48] and the m6A methyltransferase complex (Fig. 5D). However, our interactome data showed that Ezh2 predominantly interacts in mESCs with complexes involved in gene repression, which included all the DNA methyl-transferases (DNMTs), the Zfp57/Trim28 complex, which is involved in imprinting in ESCs[49], the H3K9me3 writers Ehmt1/Ehmt2 (GLP/G9A) as well as the H3K9me3 readers HUSH and HP1 complexes (Fig. 5D). Importantly, all these protein complexes showed thermal stability changes in some of their subunits in response to Ezh2 inhibition (Fig. 5D). To check if some of these interactions were compromised by the inhibition of Ezh2, we treated mESCs with GSK126 and analysed the Ezh2 interactome (Supplementary Fig. 16B). Under these conditions, we found that several DNMTs and H3K9me readers decreased their interaction with Ezh2 (Fig. 5D). Conversely, among the proteins that were found more abundant, we found the Zinc-finger protein Zfp57, Ehmt2 and all 4 subunits of the PRC2.1 complex, suggesting an alteration in the balance of PRC2 subcomplexes in response to GSK126[42]. Together, these data reveal that inhibition of Ezh2 results in thermal stability changes in the writers-erasers-readers of key epigenetic marks beyond H3K27me3, many of which interact closely with Ezh2. These results reinforce the notion of important cross-talk mechanisms between

Ezh2-H3K27me3 and DNA methylation[50,51], as well as with other histone marks[52,53]. In support of the latter, we found that GSK126 also reduced H3K4me3 levels in mESCs (Supplementary Fig. 17).

## A non-canonical function of Ezh2 in chromosome organization

In addition to its canonical function in the tri-methylation of Histone 3 at K27, Ezh2 has been reported to methylate other non-histone proteins[43]. In our data, we found numerous proteins related to cell cycle that were thermally stabilized in mESCs upon Ezh2 inhibition (Supplementary Fig. 15D). These included the key cell cycle regulator Tumor suppressor ARF (Cdkn2a), the cyclin B2 (Ccnb2), the Cohesin subunit SA-1 (Stag1) and Mki67. To investigate this further, we examined a recent report that analysed thermal stability variation across cell cycle stages in HeLa cells[15]. Remarkably, we found that stabilized proteins during mitosis in HeLa cells were also stabilized in mESCs in response to both Ezh2is with no apparent changes in abundance (Fig. 6A). Similarly, stabilized proteins in other stages (except early S) were also stabilized in our data (Fig. 6B). Moreover, our GO analyses also showed an enrichment in cytoskeleton organization and spindle positioning amongst the stabilized proteins in mESCs upon Ezh2 inhibition (Supplementary Fig. 15E). For instance, we found thermal stabilization of Abl1, a tyrosine kinase involved in cytoskeleton remodelling, Mapt, which promotes microtubule assembly, as well as cofilin Cfln1 and cortactin Cttn1 which regulate actin polymerization. Further, we found that key regulators of the mitotic spindle checkpoint, including the Mitotic spindle assembly checkpoint protein MAD1 (Mad1l1), the Mitotic checkpoint serine/threonine-protein kinase BUB1 beta (Bub1b) and the Centromere-associated proteins Cenpe, Cenpf also increased their thermal stability values (Fig. 6C). Importantly, many of these spindle proteins were found as interactors of Ezh2 in our AP-MS data (p.val 4.04E-96, Hypergeometric test), including macromolecular complexes of the spindle apparatus such as the centrasplindin complex and the chromosomal passenger complex among others (Fig. 6D). Interestingly, most of these proteins interacted less with Ezh2 in response to GSK126 (Supplementary Data 10). To rule out a potential activation of the DNA damage response in response to Ezh2 inhibitors, we checked several hallmark pathways involved in this process and did not find significant alterations (Supplementary Fig. 18). Taken together, our data suggested a potential connection between Ezh2 and cell division. However, we did not find changes in cell proliferation (Supplementary Fig. 19A), and cell cycle analysis by flow cytometry showed no differences (Supplementary Fig. 19B, C). Similarly, analysis of mitotic phases by immunofluorescence did not reveal major alterations in mESCs treated with GSK126 compared to control cells (Fig. 6E). Interestingly, we noticed that Ezh2 localized in the periphery of mitotic mESCs chromosomes (Fig. 6F), in agreement with a recent report[54]. Further, we surveyed a recent list of proteins bound to native mitotic mESCs chromosomes[54] and found an overall increase in their thermal stability values in mESCs in response to Ezh2 inhibition (Fig. 6G). Prompted by these findings, we re-examined our mitotic images data and found that Ezh2 inhibition in mESCs caused a major increase in the number of abnormal metaphases (Fig. 6H), which also exhibited overall larger areas (Fig. 6I). No differences in chromosome area, however, were found in prophase, where chromatin is normally less compacted (Supplementary Fig. 20). Therefore, our results point out a role of PRC2 in chromosome compaction, consistent with a prior report showing that *Eed−/−* (Polycomb protein EED) mESCs that lack PRC2 activity exhibit chromosomes larger in size[54].

## The perichromosomal layer protein Mki67 is a putative novel Ezh2 substrate

Among the stabilized proteins in response to Ezh2 inhibition, there was an enrichment in chromosome organization (Supplementary Fig. 14F), including numerous proteins known to be part of the perichromosomal layer of mitotic chromosomes[55] (Fig. 7A). Moreover, we found that most of these perichromosomal layer proteins were also interactors of Ezh2 in our AP-MS (Fig. 7B), suggesting a potential link between Ezh2 and this chromosomal structure. To identify potential effectors of Ezh2 in the chromosome periphery, we focused on Mki67. Mki67 primary function is in mitosis, where it is required to maintain individual chromosomes dispersed in the cytoplasm following nuclear envelope disassembly[56]. Mki67 showed increased thermal stability in response to Ezh2 inhibitors (Supplementary Data 7), and it was found to interact with Ezh2 in our data (Fig. 7B). Importantly, Mki67 protein sequence possesses the only recognition motif reported to date for Ezh2 in non-histone proteins[57]. This RKS motif resembles the canonical H3K27me3 substrate, and, so far, it has been only experimentally validated in the nuclear orphan receptor RORα where it acts as a methyl-degron for the DCAF1/DDB1/CUL4E3 ubiquitin ligase complex[57]. We noticed that Mki67 possesses up to four RKS motifs in its N-terminus (Fig. 7C). Moreover, Mki67 is one of the 29 proteins found in a previous screening of Lys-methylated proteins, which identified four Lys residues in Mki67 to be di- and tri-methylated[58] (Fig. 7C). Strikingly, two of these Kme sites fall in the RKS motif: K263 and K620. Based on these findings, we hypothesised that Ezh2 could methylate Mki67 in Lys residues containing RKS motifs. To test this, we performed an in vitro methyl-transferase assay. We incubated recombinant Mki67 with a reconstituted PRC2 complex (Ezh2, Eed, Rbbp4-7) in the presence of cold SAM and analysed the reaction by LC-MS/MS. Using DDA, we identified two lysine residues methylated in Mki67: K263 and K549. We validated these results using PRM, which confirmed the presence of K263me1 and K549me1 and revealed K263me2 at sub-stoichiometric levels (Fig. 7D and Supplementary Fig. 21). Importantly, the inhibition of Ezh2 with GSK126 prevented these methylations, and we did not identify any methylation in Mki67 alone, indicating the K263 and K549 are methylated by PRC2 in a methyltransferase dependent-manner. Moreover, in all cases, methylations prevented trypsin cleavage (internal Lys), and the methylated peptides exhibited retention times and fragmentation patterns similar to their unmodified counterparts providing further confidence in their assignments. Thus, we concluded that Ezh2 methylates Mki67 in K263, located in a RKS motif that is conserved across several species (Fig. 7C).

## Discussion

In mESCs, a handful of reports have shown that methylation regulates key factors such as Sox2[6] and Lin28[59], implying that this modification could be important for pluripotency. Our immune-purification data sets identified hundreds of novel methylated proteins in mESCs, including several factors associated with pluripotency. However, none of the previously known methylation sites in Sox2 (K119) and Lin28a (K135) were identified here. These results indicate that the methyl proteome is still under-sampled by current proteomics technologies. This is clearly evident in the case of Lys methylation, where less than 200 sites could be identified here. As a matter of fact, and compared to other modifications, few reports have systematically analysed protein methylation function and, not surprisingly, these have been exclusively focused on Arg methylation[4,9,10] as the available immunoreagents needed for purification are more selective. Here, we chose to probe the methyl-proteome from a different angle and explored the potential relationships between the methylation status of a protein and its thermal stability. This biophysical parameter changes depending on the interactions of a protein with other biomolecules, its enzymatic activity, conformational state and even post-translational modifications[14]. Given that methylation participates in all these molecular events, we reasoned that changes in the thermal stability of proteins could therefore serve as a proxy to identify potential proteins and functions controlled by methylation. Recently, Huang et al. showed that thermal stability analysis unveils shifts in overall protein stability in response to site-specific phosphorylation sites[60], albeit the extent of this finding at the proteome scale is under debate[61,62]. A major

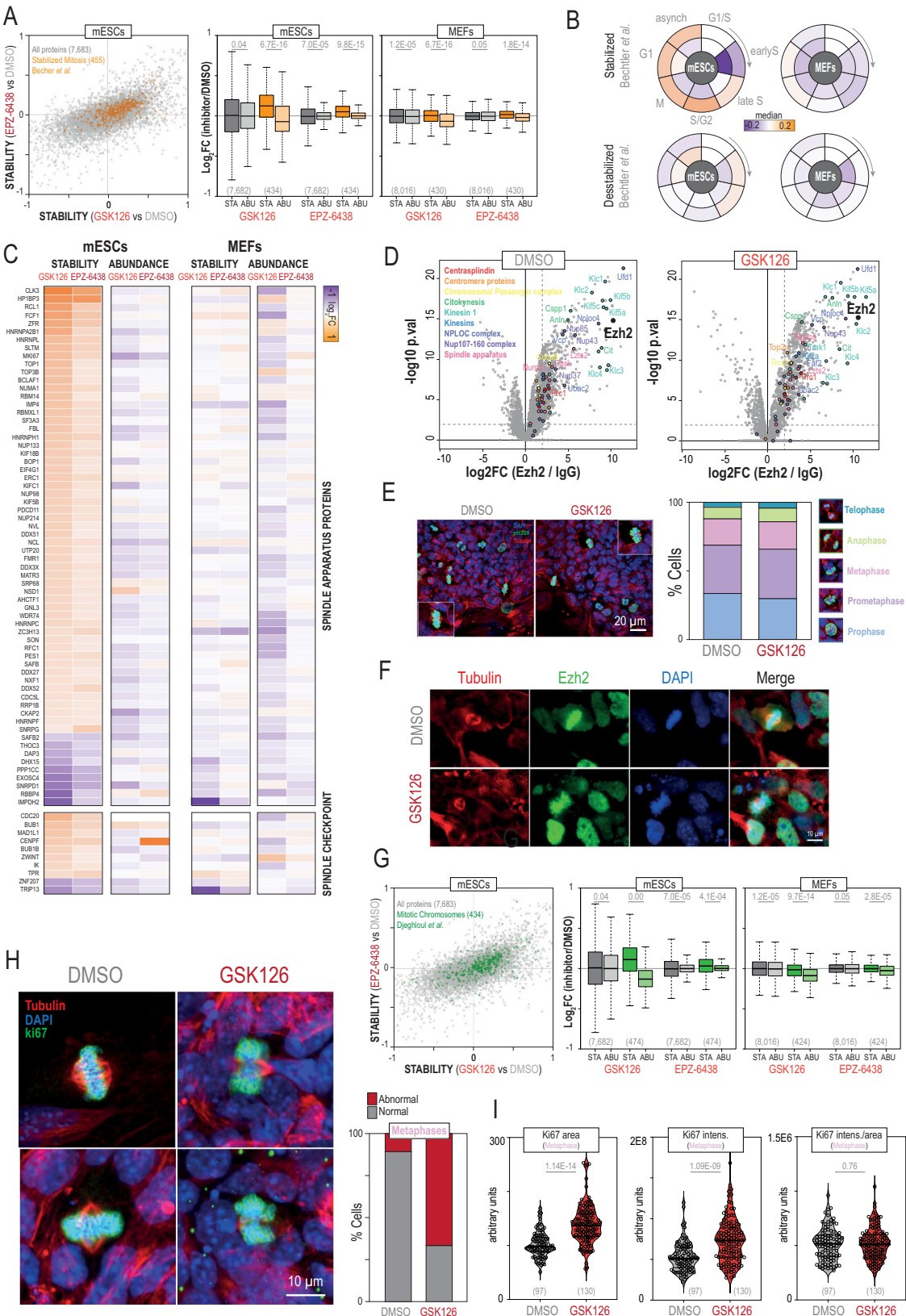

difference with our approach is the omission of a prior purification step for the subsequent thermal stability analysis of the enriched methylated peptides. Thus, it is reasonable to think that our data might be biased towards highly stoichiometric methylated residues, which, more likely, could produce detectable thermal stability differences. However, implementing an enrichment step can be methodologically

more challenging to couple with isobaric labeling (where side chains of Lys are tagged) and would also require much higher sample amounts. Furthermore, methylated peptides are significantly larger and more protonated than regular peptides, complicating their analysis using isobaric labeling and LC-MS/MS. Our strategy, on the other hand, can profile a great fraction of the proteome (≈8000 proteins) using

**Fig. 6 | Thermal stability alterations in mitotic chromosome-bound proteins reveal metaphase and chromatin compaction defects upon Ezh2 inhibition.**
**A** Scatterplot (left) showing the stability changes measured in mESCs in response to Ezh2 inhibition for 455 proteins known to be stabilized during mitosis in HeLa cells[15] (in orange); box-plots (right) showing the distributions of stability (STA) and abundance (ABU) log$_2$ ratios in mESCs and MEFs in response to Ezh2 inhibitors for proteins stabilized in mitotic HeLa cells. **B** Radial plots showing the median of the log2 ratios for stability (outer circle) and abundance (inner circle) changes in mESCs and MEFs for stabilized (upper) and destabilized (bottom) proteins across cell cycle stages in HeLa cells[15]. **C** Heatmap showing the log2 ratios for stability and abundance changes in mESCs and MEFs treated with Ezh2 inhibitors for proteins with known functions in mitotic spindle. **D** Volcano plots highlighting protein complexes with known functions in mitotic spindle identified as Ezh2 interactors in our AP-MS data. *P*-values were calculated using limma (two-sided) and adjusted for multiple testing with Benjamini-Hochberg. **E** Analysis of mitotic phases by immunofluorescence (*N* = 450 cells). **F** Localization of Ezh2 in mESCs. This experiment

was repeated twice. **G** Scatterplot (left) showing the stability changes measured in mESCs in response to Ezh2 inhibition for 434 proteins known to be bound to mitotic chromosomes in mESCs[54] (in green); box-plots (right) showing the distributions of stability (STA) and abundance (ABU) log$_2$ ratios in mESCs and MEFs in response to Ezh2 inhibitors for proteins bound to mitotic chromosomes. **H** Examples of metaphases in DMSO- and GSK126-treated mESCs (left). The number of cells with abnormal metaphases is shown on the right (*N* = 7000 cells; 100 metaphases). **I** Chromosome area was determined using Ki67, showing greater metaphases upon GSK126 treatment. Ki67 intensity normalized by area showed no differences in Ki67 recruitment to chromosomes during metaphase in response to GSK126. *P*-values were calculated with a Mann Whitney U test (two-sided). Sample sizes are shown below in parenthesis. For box-plots, median is shown; box limits indicate the 25th and 75th percentiles; whiskers extend 1.5 times the interquartile range from the 25th and 75th percentiles. Sample size in each category is shown below in parenthesis. *P*-values (**A**, **G**) are calculated using a non-parametric Wilcoxon test (two-sided). Source data are provided as a Source Data file.

routine methodologies, albeit it is limited in the detection of stability changes for proteins whose melting points fall within the temperature range used here (51–56 °C)[24].

Using inhibitors against Prmt5, we showed that hundreds of Arg methylated proteins displayed alterations in their thermal stability values, including well-known Prmt5 substrates. We confirmed G3bp2 as a Prmt5 substrate[28]. Di-methylation of G3bp2 at R468 promotes its stability through USP7-dependent de-ubiquitination, and activates de novo lipogenesis and tumorigenesis in head and neck squamous carcinoma (HNSC) cell lines[28]. Our protein abundance data, however, did not show any alterations in proteins involved in lipid metabolism, suggesting that the effect of R468me2 in mESCs may be different. Albeit methylation is a relatively small chemical moiety that does not confer any net change in the charge of the residue, differences in the methylation status of a protein can modify its thermal stability. We aimed to measure the actual impact of protein methylation on the thermal stability of a protein using an in vitro model of G3bp2 methylation without any other confounding factors. However, we did not observe significant differences in its thermal stability. Several technical reasons might explain this result. These include the presence of unmethylated G3bp2 due to incomplete methylation, the presence of mono-methylated G3bp2 owing to the distributive methyl-transferase activity of Prmt5 (Fig. 3B and Supplementary Fig. 8) and abnormal protein conformational states of the commercial recombinant G3bp2 protein used here. Improvements that more closely recapitulate the physiological conditions of Prmt5-dependent methylation of Grbp2, including the potential requirement of RNA, would help determine this question. Methylation, particularly in Arg, occurs within dense clusters of modified residues, raising the possibility that some methylated proteins might be regulated via multisite and cooperativity mechanisms, increasing the propensity of these proteins to exhibit changes in their stabilities. Nevertheless, we also found that some methylation events do not necessarily alter the thermal stability of the cognate protein. For instance, our data did not show any alteration in the thermal stability of Snrpd3, a well-known Prmt5 substrate we confirmed to be de-methylated by western blot (Supplementary Fig. 5). Moreover, Arg methylation is frequently localized in low complexity regions of proteins, a finding that we confirmed in our immunoprecipitation data sets, and is an important PTM in the formation of membrane-less organelles through LLPS mechanisms[63]. Methylation of these regions involves significant conformational changes in proteins, likely affecting several biophysical properties[64], including their thermal stabilities. Indeed, we found that a great fraction of the proteins with increased thermal stability upon Prmt5 inhibition were connected to the formation of these molecular condensates. Interestingly, Arg methylation has been found to both promote[40] and suppress[65] the formation of RNP granules. Our data support the former hypothesis as de-methylation of Prmt5 substrates resulted in fewer stress granules.

Numerous mRNAbps also increased their thermal stability, suggesting that Prmt5 might be involved in the interaction of these proteins with mRNAs. These results contrast with a recent report showing that type II (Prmt1) but not type I (Prmt5) modulate the interaction of proteins with RNAs[66].

Strikingly, Prmt1 (Type I)[30,31], Prmt5 (Type II)[30,31] and even Prmt 7 (Type III)[67] are all involved in RNA-protein interactions, acting in some cases on the same proteins. Therefore, it will be interesting to analyze potential stability changes in response to all three PRMT types. These analyses could be performed in mechanically disrupted cells treated with RNase (vs untreated), similar to the strategy described by Sridharan et al.[68], and would certainly inform on the role and cross-talk between methyl-transferases in the control of splicing and other processes involving RNA-protein interactions. However, it should be noted that proteins that undergo phase transitions might exhibit differences in their solubility[68,69]. Given that the PISA-based strategy does not discriminate between stability and solubility changes, it could be plausible that some of the stabilized proteins increase their solubility instead. In agreement with this possibility, we found that a fraction of stabilized proteins in our data has been reported before as insoluble proteins[68] (Supplementary Fig. 22).

The large number of Arg methylation sites contrasts with the low number of Lys sites identified so far, paradoxically controlled by a higher number of methyl-transferases and de-methylases. This suggests that Lys methylation networks might be more tightly regulated than Arg with regard to the number of non-histone substrates, which is probably consistent with the critical role of Lys methylation in chromatin regulation during development[70]. Ezh2 is a KMT involved in H3K27me3 but also methylates a few other non-histone proteins[43]. Unexpectedly, our data showed that Ezh2 inhibition affected the stability of hundreds of proteins. Hence, it is conceivable that many of these alterations are not directly connected to changes in methylation but instead reflections of other downstream molecular events. Although this complicates the identification of effectors regulated by Ezh2, it can, at the same time, uncover important functional information on cellular responses. Here, we showed that inhibition of Ezh2 induces several changes in the thermal stability of writers, erasers and readers of critical epigenetic marks in mESCs suggesting a cross-talk between H3K27me3 and other modifications. Whether the alterations in these proteins reflect a differential interaction with chromatin and/or histone marks is an open question. However, in support of this idea, van Mierlo et al.[71] found that naïve mESCs lacking a functional PRC2 complex (*Eed−/−*) exhibit multiple changes in chromatin-bound regulators, many of which were altered in our thermal stability data. Moreover, they showed that *Eed−/−* mESCs and GSK126 treated-mESCs showed numerous changes in several histone modifications[71].

In addition, our Ezh2 data showed thermal stability changes in proteins involved in cell cycle and mitotic chromosomes but not their

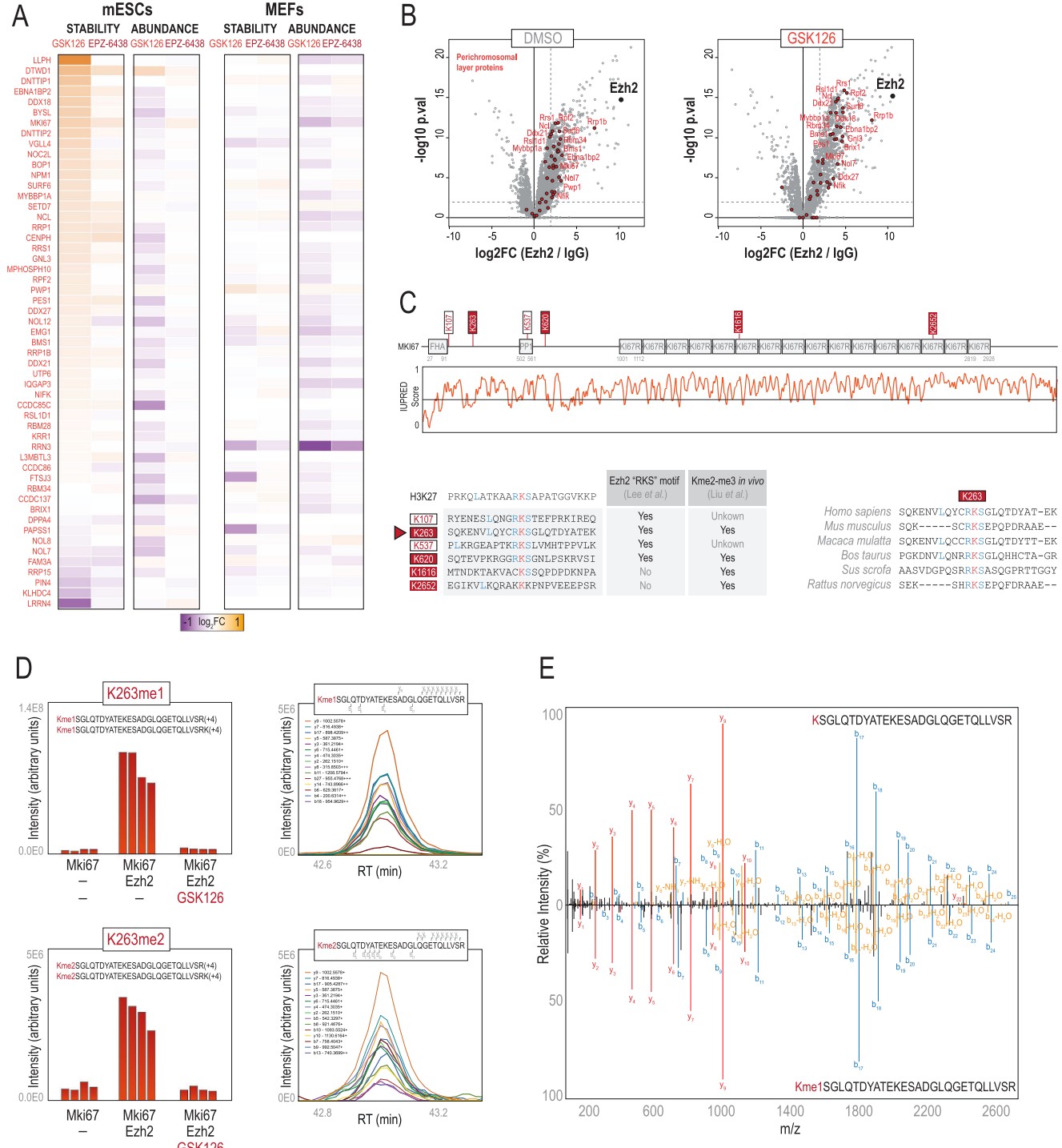

**Fig. 7 | Ezh2 inhibition in mESCs causes thermal stability alterations in proteins from the perichromsosmal layer including Mki67, which is methylated in vitro by Ezh2 in K263.** **A** Heatmap showing stability and abundance changes in mESCs and MEFs treated with Ezh2 inhibitors for proteins of the perichromosomal layer[55] identified in our AP-MS data as Ezh2 interactors (**B**). **C** Pfam domains and prediction of low complexity regions for Mki67. Lys residues in RKS motifs[57] as well as Lys residues known to be di- and tri-methylated in vivo[58] are shown. The conservation of K263 across different species is shown. **D** (left) PRM-based quantification of Kme levels (K263me1 and K263me2) in Mki67 in an in vitro methyl-transferase assay using Ezh2. The different peptide sequences used in each quantification are shown. (right) Extracted ion chromatograms (XIC) integrated in Skyline of methylated peptides showing some of the product ions used for quantification. **E** Comparison of MS/MS spectra between the mono-methylated peptide and its unmodified counterpart. Source data are provided as a Source Data file.

abundance. These results likely reflected the abnormal metaphases observed in mESCs-GSK126, highlighting the importance of thermal stability in revealing changes in the functional state of proteins. Our results are then in great agreement with a recent report showing that PRC2 is required for chromosome compaction during mitosis in mESCs[54]. Interestingly, 81% of the mitotic chromosome periphery compartment (MCPC) has at least one predicted long-disordered domain[55]. The perichromosomal layer comprises 33% of the protein mass of mitotic chromosomes[72], and Mki67 is the key protein in the formation and organization of this compartment[73]. Our data indicate

now that Ezh2 could regulate Mki67. Mki67, its interacting partners the MKI67 FHA domain-interacting nucleolar phosphoprotein (Nifk) and the Serine/threonine-protein phosphatase PP1-gamma catalytic subunit (Ppp1cc), and many other components of the MCPC, were found as Ezh2 interactors in our AP-MS data in mESCs. Further, Mki67 is one of the 16 protein readers of H3K27me3[46], and its thermal stability changed upon Ezh2 inhibition. Most importantly, Mki67 is known to be di- and tri- methylated in several Lys residues[58], some of which fall in RKS motifs, the only recognition motif reported for Ezh2 in non-histone proteins[57]. In this work, we have linked these observations and demonstrated that Ezh2 methylates Mki67 in vitro in one RKS motif (K263). Whether this methylation also acts as a methyl-degron in Mki67[57] remains to be investigated. Another possibility is that K263 methylation might be involved in the phase separation properties of Mki67 or the tethering of these intrinsically disordered C-terminal repeats to the chromosome surface[74].

In conclusion, the thermal stability analyses reported here provide novel insights into the functions and processes controlled by protein methylation. This approach represents a great addition to the proteomic toolbox identifying potential candidates of methyltransferase substrates. However, it is important to note that potential substrates would need to be validated using conventional techniques. Likewise, it is quite plausible that alternative biophysical proteomic methods, such as the Limited Proteolysis (LiP), could provide additional and more refined structural information[75] on Lys and Arg methylation function, improving our systems-view of the methylproteome. Finally, our data represent a rich resource to study the role of protein methylation in pluripotency. Based on the significant differences observed between mESCs and MEFs, our results suggest that pluripotent cells may be more susceptible to changes in protein methylation homeostasis, consistent with the importance of epigenetic control in development.

## Methods

### Cell culture
ESCs (cell lines G4, F123, E14 and V6,4) were grown in the presence of 15% serum and leukemia inhibitory factor (LIF, 103 u/mL) and cultured in P100 plates at 37 °C and 7% $CO_2$. Cells were collected by trypsinization at passage 18. MEFs were cultured in Dulbecco's modified Eagle's medium (DMEM) supplemented with 10% fetal bovine serum (FBS; Life Technologies), penicillin (100 U/ml), and streptomycin (100 mg/ml) in P150 plates at 37 °C and 5% $CO_2$. Cells were collected by trypsinization. Cells were washed with PBS and immediately frozen at −80 °C. Kidney and liver samples were obtained from mice housed at the CNIO (*Mus musculus*, C57BL/6 background, 8–10 weeks old) according to protocols approved by the CNIO-ISCIII Ethics Committee for Research and Animal Welfare (CEIyBA). The two mice used in this study were not disaggregated by sex, and both male and female were analysed. For thermal stability analysis, mESCs (G4) and MEFs were treated with inhibitors of Prmt5 (GSK3203591 and GSK3326595) and Ezh2 (GSK126 and EZP6438) which were diluted in 0.1% DMSO.

### Full proteome analysis
MEFs and ESCs were solubilized using 5% (w/v) sodium deoxycholate (SDC) in 100 mM Tris HCl pH 8.0, supplemented with 5 mM sodium butyrate, 1:100 (v/v) Halt phosphatase-protease inhibitor cocktail 100x and 1:1000 (v/v) benzonase. Samples were lysed at 90 °C during 10 minutes in agitation and then, sonicated for 2 min. Protein amount was quantified using BCA at 480 nm (Pierce Rapid Gold BCA Protein Assay Kit). Approximately 100 mg of protein were reduced (15 mM TCEP) and alkylated (15 mM chloroacetamide, CAA) for 1 h at room temperature in the dark. The excess of CAA was quenched with 10 mM dithiothreitol (DTT) for 15 min. Protein extracts were 5-fold diluted in 50 mM Tris and subsequently digested with Lys-C first (4 h at RT, 1:100 w/w) (Wako) and then trypsin (overnight at 37 °C, 1:50 w/w) (Promega).

SDC was removed by acid precipitation with TFA (2% final concentration) and clarified by centrifugation (15 min at 20,000 g). For chymotrypsin, mESCs and MEFs were lysed in 7 M urea in 50 mM Hepes, 1:1000 (v/v) of benzonase and 1:100 (v/v) of Halt phosphatase-protease inhibitor cocktail 100x. Protein concentration was measured by Qubit Protein Assay Kit. 600 μg of protein of each lysate were digested. Samples were reduced as above and diluted 8-fold in 50 mM Tris and 10 mM CaCl2. Proteins were digested twice with chymotrypsin (4 h at RT, 1:50 w/w) (o/n at RT, 1:50 w/w) (Sigma). In all cases, peptides were desalted with C18 Sep-Pack and lyophilized. MEFs and mESCs samples were pre-fractionated by high pH RP chromatography in 34–36 fractions (see below) and analyzed by LC-MS/MS using an Ultimate 3000 RSL nano LC system (Thermo Scientific) coupled to a Q Exactive HF-X mass spectrometer (Thermo Scientific) equipped with an EASY-spray ion source (Thermo Scientific). Xcalibur (Tune V2.11.0.3006) was used for instrument control. Samples were first loaded onto a trap column (100 μm i.d x 2 cm) packed with Acclaim PepMap100 C18 5 μm, and washed for 4 min at 10 μL/min with loading buffer (0.1% FA). Then, peptides were eluted from an Easy-Spray Column (75 μm i.d. x 50 cm) packed with PepMap RSLC C18 2 μm using a gradient consisting of 0.1% FA (buffer A) and 100% ACN in 0.1% FA (buffer B), with a flow rate of 250 nL/min during 60 minutes. The column was operated at a constant temperature of 45 °C. The mass spectrometer was operated in data dependent acquisition (DDA) mode. MS1 resolution was set to 60,000 resolution (m/z 200) and maximum ion injection time was 25 ms (ion target value = 3E6). The 10 most abundant isotope patterns with charge ≥ 2 and < 6 from the survey scan were selected isolated using a 2 m/z window and fragmented with HCD (normalized collision energy, NCE 27). MS2 resolution was 15,000 or 30,000 with a maximum injection time of 22 and 54 ms respectively. Ion target value was 1E5. Raw files were analysed by MaxQuant v1.6.10.43 against *Mus musculus* databases (UniProtKB: 21,982 protein sequences; Ensembl: 68,342 isoform sequences). Carbamidomethylation of cysteine was included as fixed modification and oxidation of methionine, acetylation of protein N-terminal and deamidation of asparagine and glutamine were included as variable modifications. Other parameters were set as default.

### Parallel reaction monitoring
Cell pellets were collected by trypsinization, washed with PBS and lysed using 7 M urea, 50 mM Hepes, 1:1000 (v/v) of benzonase and 1:100 (v/v) of Halt phosphatase and protease inhibitor cocktail 100x. Cell lysates were homogenized by vortex and sonication. Tissues were lysed on the Precellys 24 homogenizer (Bertin Technologies) for 10 min. Protein concentration was measured by Qubit Protein Assay Kit. 200 μg of protein of each lysate were digested. Samples were reduced (15 mM TCEP) and alkylated (15 mM CAA) for 1 h at RT in the dark. Samples were diluted 8-fold in 50 mM Tris and digested first with Lys-C (1:100 w/w, Wako) overnight at RT and then with trypsin (1:50 w/w, Promega) for 4 h at 37 °C. Digestions were stopped with 2% TFA (final concentration) and desalted on reversed-phase C18 StageTips. LC-MS/MS was performed on an Ultimate 3000 RSL nano LC system (Thermo Scientific) coupled to a Q Exactive HF-X (Thermo Scientific) equipped with an EASY-spray ion source (Thermo Scientific). Xcalibur (Tune v2.11.0.3006) was used for instrument control NanoLC was done as above but using in this case a 90 min gradient. The mass spectrometer was operated in a parallel reaction monitoring (PRM) mode using 60,000 resolution for both MS and MS/MS. Ion target values were 3e6 for MS (maximum IT = 25 ms) and 2e5 for MS/MS (maximum IT = 118 ms). Peptides were isolated using a 1.6 m/z window and fragmented with HCD (NCE 27). A precursor spectrum was interspersed every 20 PRM spectra. An isolation list containing 244 targets from methyl-transferases and de-methylases, as well as 16 peptides for normalization and RT calibration purposes were imported and divided into 2 runs. Targets were scheduled for 10 min around the expected RT

(predicted by Prosit). Data were analyzed with Skyline v20.2.0.343. A predicted spectral library was constructed using Prosit which included all 244 peptides assayed by PRM. The mProphet model was trained on all the samples based on the second-best peak model and peptides were filtered with q-value < 0.01. The intensity values from each peptide were normalized by monitoring 16 additional peptides from housekeeping proteins. Data were loaded into Prostar (v1.18.5) for further statistical analysis. Missing values were imputed using the algorithms SLSA for partially observed values and DetQuantile for values missing on an entire condition. Differential analysis was done using the empirical Bayes statistics Limma. Only proteins with 4 or more non-imputed values for at least one condition were considered for the differential analysis. Proteins above a threshold ratio of 1.5 (log2 ratio > 0.585 or < −0.585) and pval < 0.05 were defined as regulated. The FDR was estimated by Benjamini-Hochberg to be below 5%.

## Purification and analysis of methylated peptides

Four high pH pre-fractionation experiments were performed (2 x mESCs and 2 x MEFs), each one of them with 50 mg of peptides. Peptide samples were dissolved in phase A (10 mM NH₄OH) and pre-fractionated on the HPLC system at a flow rate of 500 µL/min using a Waters C18 3.5 µm 130 Å, LC column 250 × 4.6 mm, using the following gradient of phase B (10 mM NH₄OH, 90% CH₃CN): 0–35 min 35% B, 35–45 min 60%, 45–46 min 90% B. Fractions were collected every minute from 15 to 60 min. A small aliquot of 0.01% of each fraction was kept for direct LC-MS/MS analysis for full proteome analysis (see above). The remaining peptide samples were concatenated into 14 fractions and lyophilized prior immune-affinity purification using pan-specific antibodies from Cell Signaling Technology against methyl-Lys and methyl-Arg. Fractions were dissolved in 1.2 mL of 1x immunoaffinity purification buffer (Cell Signaling Technology) and subjected to two consecutive steps of methyl peptide enrichment using the PTMScan technology following manufacturer's instructions. A first purification was done from the fractions from one high pH experiment using the mono-Methyl Arginine Motif [mme-RG] Kit and the flow-through was then immuno-purified using the PTMScan Pan-Methyl Lysine Kit. In parallel, using a second HpH experiment, samples were first purified using the PTMScan [adme-R] Kit and then with the PTMScan [sdme-R] Kit. Before immune-purification, antibodies were covalently cross-linked to the beads with bis(sulfosuccinimidyl) suberate (BS3). Peptides were eluted twice with 50 µL of 0.15% TFA and desalted on reversed-phase C18 StageTips. NanoLC-MS/MS was done as above using 60 min gradients. The resolution was set to 120,000 (m/z 200) for MS1 and 60,000 (m/z 200) for MS/MS. The maximum ion injection times for the survey scan and the MS/MS scans were 25 ms and 118 ms respectively and the ion target values were set to 3e6 and 5e4, respectively for each scan mode. Raw files were analysed using MaxQuant (1.6.10.43) against a *Mus musculus* database containing the 11,655 sequences identified previously in mESCs and MEFs in the full proteome analysis. Carbamidomethylation of cysteine was included as fixed modification and oxidation of methionine, acetylation of protein N-terminal, deamidation of asparagine and glutamine, methylation of lysine (mono, di and tri) and arginine (mono and di) were included as variable modifications. Neutral losses of 31.04 Da and 45.05 Da were set for SDMA and ADMA respectively. Other parameters were set as default.

## Western blot

Cells were lysed in 2% SDS, 100 mM tris-HCl (pH 7.5) and equal amounts of protein (≈30 g) were resolved by SDS-polyacrylamide gel electrophoresis (4-12% Bis-Tris) and blotted onto the PVDF transfer membrane (Merck Millipore). Membranes were stained with Ponceau and washed in PBS-Tween 0.01%. Blots were blocked in 5% non-fat milk in PBS-Tween at RT for 30 min, followed by incubation o/n at 4 °C with the primary antibody (SDMA, Cell Signaling Technology, 13222, 1:1000

dilution; SmD3, Sigma, HPA001170, 1:1000 dilution; α-Tubulin, Cell Signaling Technology, 2125, 1:1000 dilution; Prmt5, Santa Cruz Biotechnology, sc-376937, 1:1000 dilution; Tri-Methyl-Histone H3 K27, Cell Signaling Technology, 9733, 1:1000 dilution, Histone H3, Cell Signaling Technology, 4499, 1:2000 dilution; Ezh2, Cell Signaling Technology, 5246, 1:1000 dilution; Tri-Methyl-Histone H3 K4, Cell Signaling Technology, 9725, 1:1000 dilution; all prepared in blocking buffer) and the with a secondary antibody at 4 °C for 5 h (goat α-rabbit IgG-680; A-21109, Invitrogen, 1:5000 dilution; goat α-mouse IgG-680; A21057, Invitrogen, 1:5000 dilution). Blots were imaged with the Odyssey Infrared Imaging System (LI-COR, NE, USA).

## Proteome Integral Solubility Analysis

mESCs and MEFs were treated with DMSO or Prmt5 and Ezh2 inhibitors at the concentrations and times indicated in the text above. Cells were collected by trypsinization and resuspended in PBS with protease inhibitors. Then, cell suspensions were distributed in 6 aliquots into a PCR-well plate. All samples were initially equilibrated at RT for 3 min and then subjected to 51 °C, 52 °C, 53 °C, 54 °C, 55 °C, and 56 °C for 3 min in a Verity Applied Biosystems thermocycler. Then, cells were equilibrated for 3 min at RT and lysed with 0.5% NP-40 and Halt phosphatase-protease inhibitor cocktail at 4 °C for 30 min. Temperature gradient samples were pooled in one tube and proteins sedimented at 100,000 g for 20 minutes at 4 °C using a Beckman ultracentrifuge (TLA 120.2 rotor). To measure total protein abundance, an additional aliquot was lysed with 5% SDS in 20 mM TEAB at 90 °C for 20 min. Protein concentration was quantified by BCA. Proteins were reduced, alkylated and digested with trypsin. Samples were desalted on reversed-phase C18 StageTips. 60-70 µg of peptides were labelled using the TMTpro (Thermo) following manufacturer's instructions. Labelled samples were mixed, cleaned-up with C18 Sep-Pack and dissolved in 10 mM of NH₄OH for subsequent fractionation by high pH reversed phase chromatography into 35 fractions as described earlier. NanoLC-MS/MS was done as above using 60 min gradients using a normalized collision energy of 32. MS2 resolution was 45,000, maximum injection time of 86 ms (ion target value = 1E5). Raw files were analysed using MaxQuant (1.6.10.43) against a *Mus musculus* database (UniProtKB, 21,982 sequences). Sample quantification type was set to TMT. Other parameters were set as default. Reporter intensities were extracted from the 'proteinGroups.txt' table and loaded into Prostar (v1.1835). Briefly, a global normalization of log2-transformed intensities across samples was performed using the LOESS function. Differential analysis was performed using the empirical Bayes statistics *limma*. Proteins above a threshold log2 ratio > 0.1 or < −0.1 and pval < 0.01 were defined as regulated. The FDR was estimated by Benjamini-Hochberg to be below 5% in all data sets.

## Functional enrichment

Enrichment analyses of different proteins subsets were performed using PANTHER database and adjusted by FDR. Alternatively, Gene Set Enrichment Analysis (v 4.0.2) was also employed using the pre-ranked algorithm and the log2 ratios as input files and adjusted by FDR.

## Immunofluorescence

Cells were treated with DMSO and 50 nM of Prmt5 inhibitor GSK595 during 2 days. Cells were treated with or without arsenite (0.5 mM for 30 min). Cells were washed twice with PBS and fixed in 4% paraformaldehyde. Then, cells were washed once with PBS and permeabilized by 0.5% Triton at room temperature. Primary antibodies were incubated overnight at 4 °C. Primary antibodies used were: FBL, (Cell Signaling, 2639; 1:200 dilution); eIF4e (Invitrogen, MA1-089, 1:200 dilution). All secondary antibodies were used at 1:200 (goat α-rabbit IgG-568, ThermoFisher, A-11036; goat α-mouse IgG-488, Thermo-Fisher, A-11001) and DAPI for 30 minutes at 25 °C. Images were

acquired with a 20 x dry objective (HC PL FLUO 0.50 NA) using a Thunder Imager Leica wide field microscope. Image analysis was performed by using a custom-made ruleset to identify and quantify Nucleus, cytoplasm and stress/nucleoli Definens Developer XD v2.0 (Definiens). To study the proliferation and mitosis by immuno-fluorescence, mESCs were seeded in human recombinant laminin 511 (Biolamina) coated 96 Well Microplate, PS, µClear®, Chimney Well (Greiner) and treated for 48 h with GSK126. Then, they were fixed by adding to each well the same volume of 8% buffered paraformaldehyde (PFA) than the culture media, for 10 min at room temperature. Cells were washed twice with PBS and they were incubated with PBS-0.5%-Triton-X100-0.05% SDS for 10 min at room temperature for permeabilization. Cells were then blocked with 5% normal goat serum-3% BSA- 0.05% Tween-20 during 2 h at room temperature followed by primary antibody incubation overnight at 4 °C. Cells were washed twice with PBS, incubated with 100 ng/ml of DAPI for DNA staining and analyzed in an Opera Phenix® Plus High-Content Screening System (Perkin Elmer), were at least, ten pictures per well were taken. The following antibodies were used:, Anti-Ki67 antibody [SP6] (Abcam, ab16667, 1:1000 dilution), Anti-Tubulin [DM1A] (Sigma, T9026, 1:1000 dilution). Secondary antibodies were purchased from Molecular Probes (Invitrogen) (goat α-rabbit IgG-488, A-11034, dilution 1:400; donkey α-mouse IgG-647, A-31571, dilution 1:400). Image analysis was performed using ImageJ (v1.52a) software.

### Flow cytometry assay

Cell cycle analysis was performed by flow cytometry with propidium iodide staining of DNA. Cells were washed with cold phosphate-buffered saline three times, fixed in cold 70% ethanol overnight at 4 °C and washed with cold phosphate-buffered saline two times. Cells were treated with Ribonuclease (Qiagen, 19101) to ensure that only DNA was stained. DNA was stained by addition of 50 µg/mL propidium iodide (Sigma, P4170). Samples were run on a FACS CANTO II flow cytometer (BD Biosciences, San Jose CA). We use pulse height and area to exclude aggregates from the analysis, at least 10,000 single events were collected. All data was analysed in FlowJo v10 applying Watson cell cycle model to quantify the different cell cycle phases

### In vitro methyl-transferases assays

The Prmt5 in vitro methylation assays were performed in 50 mM HEPES, 50 mM NaCl, 1 mM EDTA, 5 mM DTT and 0.2 mM S-adenosylmethionine. 4 µg of G3bp2 (abcam, AB123193) was incubated with 0.6 µL of active human recombinant Prmt5-Mep50 complex for 15 h (Sigma, SRP0145), in the presence or absence of GSK595. The Ezh2 in vitro methylation assays were performed in 50 mM Tris HCl, 50 mM MgCl2, 4 mM DTT and 20 µM S-adenosylmethionine. 1 µg of Mki67 (Origene, TP710117) was incubated with 1.5 µg of active human recombinant PRC2 complex (Sigma, SRP0134) for 6 h, in the presence or absence of GSK126. Reactions were quenched with 7 M urea in 50 mM HEPES, proteins were reduced-alkylated as above, diluted 8-fold in 50 mM HEPES and trypsin-digested. LC-MS/MS was performed on an Ultimate 3000 RSL nano LC system (Thermo Scientific) coupled to a Exploris 480 (Thermo Scientific) equipped with an EASY-spray ion source (Thermo Scientific). Xcalibur (Tune v4.0) was used for instrument control. NanoLC was done as above but using a 60 min gradient. The mass spectrometer was operated in a parallel reaction monitoring (PRM) mode using 60,000 resolution for both MS and MS/MS. Ion target values were 3e6 for MS (maximum IT = 25 ms) and 2e5 for MS/MS (maximum IT = 118 ms). Peptides were isolated using a 1.6 m/z window and fragmented with HCD (NCE 27). A precursor spectrum was interspersed every 16 PRM spectra. Targets were scheduled for 10 minutes around the expected RT (predicted by Prosit). Data were analyzed with Skyline v20.2.0.343.

### Immunoprecipitation of Ezh2

Cells (mESCs) were treated with DMSO or 2 µM GSK126 for 2 days. Cells were washed with cold phosphate-buffered saline three times and lysed for 30 min on ice in 1 mL NP40 lysis buffer (150 mM NaCl, 50 mM Tris pH 8.0, 1 mM EDTA and 1% NP-40) supplemented Halt phosphatase-protease inhibitor cocktail. Cell extracts were cleared by centrifugation at 16,000 g for 10 min at 4 °C. Protein amount was quantified using BCA at 480 nm (Pierce Rapid Gold BCA Protein Assay Kit). 500 µg of protein were incubated with 1 µg of IgG (Cell Signaling Technology, 3678) or Ezh2 antibody (Cell Signaling Technology, 5246) overnight at 4 °C on a rotating wheel, washed with NP40 lysis buffer three times and incubated for 4 h at 4 °C on a rotating wheel with 15 µL of Protein A dynabeads (10001D, Life Technologies). Dynabeads were washed three times with NP40 lysis buffer and twice with 100 mM ammonium bicarbonate. Proteins were on-bead digested with 10 µL of digestion buffer (10 ng/µL of trypsin in 100 mM ammonium bicarbonate) at 37 °C overnight. LC-MS/MS was performed on an Ultimate 3000 RSL nano LC system (Thermo Scientific) coupled to a Exploris 480 (Thermo Scientific) equipped with an EASY-spray ion source (Thermo Scientific). Xcalibur (Tune v4.0) was used for instrument control. NanoLC was done as above but using a 60 min gradient. The mass spectrometer was operated in a data-independent acquisition (DIA) mode using 60,000 precursor resolution and 30,000 fragment resolution. Peptides were fragmented using HCD with a normalized collision energy of 27 and assuming a default charge state of +2. The ion target values were 3e6 for Full MS (maximum IT of 25 ms) and 1e6 for DIA MS/MS (maximum IT of 54 ms). 4 m/z precursor isolation windows were used in a staggered-window pattern from 400.4 to 1000.7 m/z. A precursor spectrum was interspersed every 76 DIA spectra. The scan range of the precursor spectra was 390–1000 m/z. Data were analysed by DIA-NN v.1.7.16 against a *Mus musculus* database (UniProtKB, 21,982 sequences). Other parameters were set as default. Reporter intensities were loaded into Prostar v1.18.5. Briefly, a global normalization of log2-transformed intensities across samples was performed using the LOESS function. Differential analysis was performed using the empirical Bayes statistics limma. Proteins above a threshold log2 ratio > 0.1 or < −0.1 and pval < 0.01 were defined as interactors. The FDR was estimated by Benjamini-Hochberg to be below 5% in all data sets.

### Reporting summary

Further information on research design is available in the Nature Portfolio Reporting Summary linked to this article.

## Data availability

The mass spectrometry proteomics data generated in this study have been deposited in the ProteomeXchange Consortium via the PRIDE partner repository under accession code PXD038939. Mass spectrometry data were searched against a *Mus musculus* database (UniProtKB, 21,982 sequences) Source data are provided with this paper.

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

## Acknowledgements

We thank all members of the CNIO Proteomics Unit for discussions, the CNIO Flow Cytometry Unit for flow cytometry support, Cyan Lynch for sharing reagents and Ana Martinez-Val for support with data analysis. This work was supported by SAF2016-74962-R (MINECO) and the European Union Horizon 2020 program INFRAIA project EPIC-XS (project 823839). The CNIO Proteomics Unit belongs to ProteoRed, PRB3- ISCIII, supported by grant PT17/0019/0005. J.M. is supported by the Ikerbasque Programme, Basque Foundation for Science. O.F.-C. is supported by grants from the Spanish Ministry of Science, Innovation and Universities (PID2021-128722OB-I00, co-financed with European FEDER funds) and the Spanish Association Against Cancer (AECC; PROYE20101FERN). M.M. lab was supported by grants from MINECO (PID2021-128726 and PDC2022-133408-I00), and Comunidad de Madrid (Y2020/BIO-6519 and S2022/BMD-7437).

## Author contributions

C.S. performed all experiments and analysed the data. J.S., P.X-E., F.G., and E.Z. contributed to the proteomic analyses. V.L. and O.F.-C. contributed to the stress granules assays. P.P. and S.O. assisted with cell culture. B.H. and M. M. performed all cell cycle analyses. D.M. analyzed immunofluorescence microscopy data. C.S. and J.M. planned all the experiments and wrote the manuscript. All authors contributed to and approved the final version of the manuscript.

## Competing interests

The authors declare no competing interests.

## Additional information

**Supplementary information** The online version contains Supplementary material available at https://doi.org/10.1038/s41467-023-38863-1.

 