## [Peer Review File · Nature Communications]

REVIEWER COMMENTS

Reviewer #1 (Remarks to the Author):

In this manuscript, the authors set out to develop new technology of thermal stability analysis of methylated proteins and understand how protein arginine methylation and protein lysine methylation influence different cell types. They present as stated “a rich resource to study the role of protein methylation in pluripotency and show that mESCs are particularly sensitive to perturbation in the homeostasis of protein methylation networks”. Overall, this is a very comprehensive study with a many interesting, well controlled techniques studying questions important for multiple fields. However, I’m concerned that there is too much information in the manuscript that may get lost. I think the authors should consider splitting this study into one targeting PRMT5 and arginine methylation and the other targeting EZH2 and lysine methylation. Honestly, no new work would be necessary, there is a tremendous amount already in the paper and the reception will likely be much easier. It will be hard for investigators in either field to dissect the important information as presented. Regardless, this work is important to be published, but I suggest the authors address the following points:

The comparisons between mESCs and MEFs are very interesting and important for the field. For instance, the difference in magnitude between EZH2 inhibition in the two cell types is potentially critical for understanding the enzymes’ role. However, the authors do not address or test biological consequences of any of these downstream players. For instance, but not limited to, what biological role could G3bp2 methylation be playing? Ultimately, why might protein methylation be relevant for pluripotency? And ultimately that conclusion from this work is just correlative with mESCs and MEFs. This should be addressed in the text, but ultimately this paper is just too long to make these points clearly.

The conclusion that Rme2s/SDMA alters the stability of proteins is very interesting, and an important observation for future work. Open questions that could be explored, but are likely beyond this manuscript, are what the role of nucleic acids are in the methylation dependent stability. For instance, the G3bp2 (a stress granule protein) in vitro stability assay could be influenced by the presence of RNA (e.g. by permeabilizing cells and treating with nuclease prior to heating), this could be an important followup experiment to do now or at least note in the discussion (expand the section in lines 412-414 to say how this might occur, what might be the ramifications). Or using the proteomic “OOPS” technique as in the cited Maniaci et al paper (DOI: 0.3389/fmolb.2021.688973)

It is not clear how the authors determined that “both compounds significantly increased the stability of Arg methylated proteins (Figure 2E)” (line 144). Figure 1E shows that all proteins had altered stability. The right hand panels don’t shown any statistical tests for the comparison between the total proteome and the Rme proteins.

The authors add new insight into Rme and Kme sites through PTMScan analysis. However, many other studies in recent years have done similar work in a range of cell types, broadening the scope. I suggest the authors consider the evidence (such as exploration of types of targets, sequences, HIC fractionation, ETD (for high charge states) and CID fractionation, etc) in line with their studies and also cite the following relevant papers:

Doi: 10.1016/j.ccell.2019.05.014

Doi: 10.1016/j.isci.2021.102971

Doi: 10.1016/j.ccell.2019.07.003

Doi: 10.1038/s41467-021-21963-1

Minor critique:

- Figure 1A is not arranged to easily discern ESCs vs MEFs. I suggest considering another arrangement, maybe two separate plots, rather than relying on the subtle color differences
- Similarly, Fig 1C would be clearer if arranged vertically so reading the MTase names would be simpler (like in all the subsequent figures with vertical heatmaps)
- Line 138 has a citation for the association with Wdr77 (#27) that is about *C. elegans* PRMT5 which does not have an associated cofactor. Change the citation to another report or a review.
- On line 156, it is noted that Alyref and Snrpn interact with PRMT5. Citations are necessary for these specific claims.
- On line 170, it is not clear what "...R468me is highly stoichiometric" means. This use of the word is also found in line 383. This is not a typical use of the word. Please define – does it mean that the residue is highly modified? Or what?
- On line 195, what does "we did not see such trend for non-polyadenylated RNA..." mean? I suggest this be defined more clearly, as it could be an important point. But what are the authors addressing? rRNAs? snoRNAs? snRNAs? tRNAs? Histone genes? Define it more clearly.

Reviewer #2 (Remarks to the Author):

The study by Sayago et al represents an impressive effort in mapping the effect of protein methylation in mouse embryonic stem cells and comparing that to mouse embryonic fibroblasts. It is shown convincingly that inhibition of demethylase activity changes the thermal stability of several hundred proteins thus likely modulating their function. Follow up in vitro assays further cement these findings by identifying a novel PRMT5 substrate. The complementarity of the method to standard abundance proteomics is also shown by lack of correlation between changes of the two measurements. The data reveals multiple proteins related to stress granule formation and it is shown that arsenite treatment in the presence of the inhibitor modulates the extent of granule formation. Further work with EZH2 inhibitors reveals links to mitotic proteins and reveals a role of EZH2 in mitotic chromosomes.

This is impressive work and will be widely used in the proteins methylation field I believe. I recommend publication following minor changes.

Comments:

Figure 5F,H,G plots: are these multiple replicates? If yes, it would be better to analyze with NPARC approach to deduce p-values for differences, rather than summing up.

“GSK595 alone however did not promote the formation of stress granules but led to alterations in nucleolar fragmentation” could you discuss whether any nucleolar proteins are specifically affected in stability?

For discussion section: PISA assay does not differentiate between stability and solubility changes, so it is conceivable that some proteins linked to LLPS are changing more in solubility than stability (see eg here: PMID: 35864335, PMID: 30858367), this would be good to mention in the discussion section.

Reviewer #3 (Remarks to the Author):

The manuscript titled "DECODING PROTEIN METHYLATION FUNCTION WITH THERMAL STABILITY ANALYSIS" elaborated versatile features of protein methylation and identified several substrates of Prmt5 and Ezh2 in mESCs and MEFs mainly through analyzing the alterations in thermal stability of proteins. Although the approach of decoding protein methylation function proposed in this article has a promising application prospect, the proteins analyzed here are still not comprehensive and lack sufficient biochemical experimental data. In addition, multiple typos are found in the current version. Some specific concerns are outlined below:

1. This manuscript identified G3bp2 as a novel Prmt5 substrate, however, an article titled "USP7- and PRMT5-dependent G3BP2 stabilization drives de novo lipogenesis and tumorigenesis of HNSC" has recently been published on the journal Cell death & disease and this article elaborated the function of Prmt5-mediated G3bp2-R468me2. Suggest to carefully read this article and make appropriate adjustments of the relevant part in your manuscript.
2. Suggest to provide a reasonable explanation of the data about mono-methylation in R468 of G3bp2 in figure 3B because the level of R468me is significantly elevated in the presence of GSK595.
3. Although the manuscript gave the reasons why the thermal stability of G3bp2 only has small difference, the author should summarize the advantages and disadvantages of this approach and give some perspectives on the improvement of this approach.
4. Many protein names in the article appear in the form of abbreviations, which creates obstacles to the reading of the manuscript, it is recommended to give the full name and the corresponding abbreviation when the protein first appears.
5. There are many "'s" in the manuscript and suggest to change them to "s". Line 284 and 289, change "DNMT's" to "DNMTs"; Line 304, change "Ezh2i's" to "Ezh2is".
6. In the section of "A non-canonical function of Ezh2 in chromosome organization", the author should first conclude the canonical function of Ezh2 and then show the non-canonical function of Ezh2 in chromosome organization to make this part more clearly.
7. Suggest to define a relationship between the changes of thermal stability and the function of the protein methylation. When using inhibitors of Prmt5 and Ezh2, some proteins with slight alterations of thermal stability compared to their methylated forms are defined as substrates and exhibit critical functions. However, it is unreasonable to conclude that proteins with changes of thermal stability regardless of the extent of the changes are all substrates or have significant functions when they are methylated.
8. The proteins can be methylated presented in the manuscript are all increased in thermal stability values when using inhibitors of Prmt5 and Ezh2, however, this is not comprehensive enough. The changes of thermal stability include both increases and decreases, so suggest to analyze some proteins with decreased thermal stability values and clarify the relationship of such changes and function.

The items below are several mistakes which need revising:

1. Line 31: change "others" to "other".
2. Line 237: change "SMDA" to "SDMA".
3. Line 290: change "its" to "their".
4. Line 337-338: delete "in" because this word appears twice in a row.
5. Line 345: change "possess" to "possessed" or "possesses".

6.Line 404: change "involve" to "involves".

7.Line 419: change "Ezh2 is a KMTs" to "Ezh2 is a KMT".

8.Line 434: change "abundances" to "abundance" because abundance belongs to uncountable nouns.

9.Line 446: delete "that" because this word appears twice in a row.

10.Line 752: change "2%SDS" to "2% SDS".

11.Line 762: change "4C" to "4°C".

It is recommended to go over the manuscript carefully again to avoid grammatical errors. Moreover, some sentences do not conform to the characteristics of English writing. For example, the sentence in line 436-437. Suggest to change this sentence to "Our results are then in great agreement with a recent report showing that PRC2 is required for chromosome compaction during mitosis in mESCs". In summary, the language of the manuscript needs to be polished.

Reviewer #1 (Remarks to the Author):

In this manuscript, the authors set out to develop new technology of thermal stability analysis of methylated proteins and understand how protein arginine methylation and protein lysine methylation influence different cell types. They present as stated “a rich resource to study the role of protein methylation in pluripotency and show that mESCs are particularly sensitive to perturbation in the homeostasis of protein methylation networks”. Overall, this is a very comprehensive study with a many interesting, well controlled techniques studying questions important for multiple fields. However, I’m concerned that there is too much information in the manuscript that may get lost. I think the authors should consider splitting this study into one targeting PRMT5 and arginine methylation and the other targeting EZH2 and lysine methylation. Honestly, no new work would be necessary, there is a tremendous amount already in the paper and the reception will likely be much easier. It will be hard for investigators in either field to dissect the important information as presented

We thank Reviewer#1 for the supportive comments on our work. The reviewer's opinion that the manuscript currently has an abundant amount of data that could potentially be overlooked is a viewpoint we somehow concur with. However, since this is the first demonstration of studying protein methylation function through thermal stability analysis, we deemed a thorough analysis necessary.

In this regard, we considered important to study both Arg and Lys methylation (targeting Prmt5 and Ezh2, respectively) because they seem to regulate different biological processes. In each case, we (i) profiled global thermal stability changes to obtain an overview of their potential downstream effects, (ii) identified novel methylated substrates (i.e. G3bp2 and Mki67) and (iii) showed their involvement in regulating specific biological processes (i.e. stress granules and mitotic chromosomes condensation). In addition, comparing our findings in pluripotent mESCs with those of differentiated MEFs was important, given the profound biological differences existing between these cell types, which was evidenced also here for the control of protein methylation. Lastly, mapping methylated proteins in each cell type was also crucial for interpretation of the thermal stability data. We are pleased to note that the reviewer acknowledges the abundance of data in our manuscript and suggests that it could be split into two separate studies. However, following the editor’s suggestion, we will maintain the main structure and sections of the manuscript.

Nevertheless, we have revised all the data included in the original submission and, to improve the accessibility of our work, we have decided to simplify our expression analysis of KMTs, KDMs, and RMTs which may be less relevant considering the main focus of our study on thermal stability. Specifically, we have removed Figure 1B (re-analysis of RNAseq), Suppl. Figures A, B, D, E and moved Figure 1C to Suppl. Figure 1B (in the revised version). We have also reduced the length of the section “The non-histone methyl proteomes of mESCs and MEFs” accordingly (from 735 to 475 words). In addition, we have decided to exclude the paragraph discussing the possible link between Ezh2 and spindle proteins (Pages 13-14, lines 450-461 in the original submission), along with its Suppl. Figure 22. This decision was made as we believe that this information does not significantly contribute to the main message of our study, and without further experimental data, is perhaps too speculative.

Regardless, this work is important to be published, but I suggest the authors address the following points:

The comparisons between mESCs and MEFs are very interesting and important for the field. For instance, the difference in magnitude between EZH2 inhibition in the two cell types is potentially critical for understanding the enzymes’ role. However, the authors do not address or test biological consequences of any of these downstream players. For instance, but not limited to, what biological role could G3bp2 methylation be playing? Ultimately, why might protein methylation be relevant for pluripotency? And ultimately that conclusion from this work is just correlative with mESCs and MEFs. This should be addressed in the text, but ultimately this paper is just too long to make these points clearly.

We appreciate the reviewer's feedback regarding the biological consequences of some of the targets identified in our work. However, we would like to clarify that our aim was not to provide a detailed biochemical characterization of these targets. Instead, our goal was to introduce a novel and complementary approach to immune-purification, based on thermal stability analysis, to study protein methylation in a large-scale manner. Through our experiments, we analyzed both forms of protein methylation, Arg and Lys, and

showed that changes in thermal stability inform on different biological functions controlled by this modification, including the regulation of stress granules and the compaction of mitotic chromosomes.

While we understand the interest in gaining mechanistic insights into the actual molecular consequences of protein methylation for some of these targets, we feel that it may deviate from the message we aimed to convey in our study. Nonetheless, we hope that our data sets will be used by other researchers to explore the biological significance of some of these targets. In the case of G3bp2, Wang et al. (PMID: 36878903) recently identified a novel mechanism by which PRMT5-mediated G3BP2-R468me2 enhances the binding to USP7, leading to the deubiquitination and stabilization of G3BP2. This stabilization ultimately activates ACLY and stimulates de novo lipogenesis and tumorigenesis (see comment #1 of Reviewer#3). We have included this reference and modified the text accordingly.

We also agree with this reviewer's opinion that the actual relevance of protein methylation in pluripotency remains to be demonstrated. However, our work provides several lines of evidence that support our observation that pluripotent mESCs are more sensitive to perturbations in protein methylation homeostasis than differentiated MEFs. In all the experiments conducted here, we observed a far greater number of changes (4-5 times more), both in terms of thermal stability and protein abundance, for the two inhibitors used to target Prmt5 (Supplementary Figure 7) and Ezh2 (Supplementary Figure 13). The reason behind this observation will require further investigation, but it is worth noting that mESCs showed a higher expression of most methyl-transferases and de-methylases than MEFs (Figure 1A), with Prmt5 and Ezh2 among the most differential proteins. This could be explained by the importance of epigenetic control in pluripotent cells, as Prmt5 has been shown to be essential for early mouse development and to maintain pluripotency (PMID: 21159818), while deposition of H3K27me3 by Ezh2 (as part of the PRC2 complex) is important for reprogramming (PMID: 22388813) and is a key epigenetic mark in development (PMID: 16630819). Analyzing the impact of targeting other methyl-transferases and de-methylases, particularly those expressed at lower and similar levels between pluripotent and differentiated cells may shed some light on this question. Nevertheless, to avoid over-interpretation of our data, we have toned down our statement that "mESCs are particularly sensitive to perturbation in the homeostasis of protein methylation networks" to a more cautious statement (Page 14, lines 473-476 in the revised manuscript file).

The conclusion that Rme2s/SDMA alters the stability of proteins is very interesting, and an important observation for future work. Open questions that could be explored, but are likely beyond this manuscript, are what the role of nucleic acids are in the methylation dependent stability. For instance, the G3bp2 (a stress granule protein) in vitro stability assay could be influenced by the presence of RNA (e.g. by permeabilizing cells and treating with nuclease prior to heating), this could be an important followup experiment to do now or at least note in the discussion (expand the section in lines 412-414 to say how this might occur, what might be the ramifications). Or using the proteomic "OOPS" technique as in the cited Maniaci et al paper (DOI: 0.3389/fmolb.2021.688973)

The role of Arg-methylation in regulating the interaction of proteins and RNA is certainly an interesting question. Our results reveal multiple RNAbps that showed alterations in their thermal stability values. Whether these changes reflect the assembly or disassembly of RNA-protein complexes cannot be inferred from our current results as Arg methylation has been shown to have both positive and negative effects in regulating these molecular interactions. The experiment suggested by the reviewer (pre-treatment with nucleases before the PISA) could indeed shed some light on this matter. Interestingly, Sridharan and colleagues recently explored the role of phosphorylation in the regulation of biomolecular condensates (PMID: 35864335). These authors measured protein solubility changes from mechanically disrupted HeLa cells after preserving or digesting cellular RNA. As the interplay between Arg methylation and phosphorylation is known to regulate phase separation and RNP granule dynamics (PMID: 30587571), it would be insightful to perform further analyses similar to those conducted by Sridharan (PMID: 33856219) that consider the cross-talk between these two modifications. Such investigations will provide valuable insights into the underlying mechanisms responsible for the formation and dissolution of biomolecular condensates. In the revised version, we have discussed this possibility (Page 13, lines 421-424).

It is not clear how the authors determined that “both compounds significantly increased the stability of Arg methylated proteins (Figure 2E)” (line 144). Figure 1E shows that all proteins had altered stability. The right hand panels don't shown any statistical tests for the comparison between the total proteome and the Rme proteins.

We acknowledge that our statement that “both compounds significantly increased the stability of Arg methylated proteins (Figure 2E)” was probably inaccurate and requires clarification. Figure 2E shows the observed changes in thermal stability in GSK591 and GSK595 for the 8,104 proteins detected in mESCs, including 1,294 Arg-methylated proteins. Statistical analysis revealed that 545 and 661 proteins increased and decreased in stability, respectively. Among these, 151 proteins were found to be methylated in Arg residues in our PTMscan analyses, with a clear bias towards proteins that increased thermal stability (95) compared to those that decreased (56). In agreement with this observation, the actual thermal stability ratios for all 1,294 Arg-methylated proteins (box-plots in the right-hand panel) also showed a trend towards increased stability. We assessed the statistical difference in thermal stability against the observed in abundance levels to demonstrate that this increase was not a consequence of higher expression level. Certainly, the statistical assessment against all the proteins is also needed and we want to thank the reviewer for bringing this to our attention. We have now performed all the corresponding statistical analyses and included the resulting p.val in Figure 2E. Our findings indicate that, upon Prmt5 inhibition, Arg methylated proteins (including ADMA and SDMA) showed an overall higher stability than the total proteome.

We would like to clarify that our intention was not to imply that “only” Arg-methylated proteins had altered stability. As noted by the reviewer, there was also a large fraction of proteins that increased stability that were not Arg-methylated (at least, they were not found as such in our PTMscan data). It is highly probable that the proteins that displayed increased stability but were not Arg-methylated underwent certain molecular re-arrangements that were not directly associated with changes in methylation (see Discussion, Lines 420-424 in the original manuscript file). A functional enrichment analysis of all the proteins (methylated and non-methylated) that were found with altered thermal stability upon Prmt5 can be found in Suppl. Table 8 and revealed alterations in several biological processes. We have updated this statement to clearly state that not all Arg-methylated proteins displayed thermal stability changes (Page 4, lines 127-128 and lines 132-133 in the revised version of the manuscript file). We have also changed the header of this section from “Arg methylated proteins, including known-substrates, show alterations in thermal stability upon inhibition of Prmt5 in mESCs.” to “Prmt5 inhibition in mESCs leads to thermal stability changes of numerous Arg-methylated proteins, including known-substrates.” which more accurately reflects our results. (Page 4, lines 97-98)

The authors add new insight into Rme and Kme sites through PTMScan analysis. However, many other studies in recent years have done similar work in a range of cell types, broadening the scope. I suggest the authors consider the evidence (such as exploration of types of targets, sequences, HIC fractionation, ETD (for high charge states) and CID fractionation, etc) in line with their studies and also cite the following relevant papers:

Doi:10.1016/j.ccell.2019.05.014

Doi:10.1016/j.isci.2021.102971

Doi:10.1016/j.ccell.2019.07.003

Doi:10.1038/s41467-021-21963-1

Our catalogues of methylated sites were an important element to interpret the thermal stability results. Certainly, these data sets offer a valuable resource to understand the molecular determinants of protein methylation and might instruct future technological improvements to profile this modification more comprehensively. In our work, we just aimed to highlight some features observed in our data, most of which have been indeed reported by others before. The authors apologize to colleagues whose work was not cited in our original submission owing to space limitations or our oversight. Nevertheless, we acknowledge the value of the studies mentioned by the reviewer regarding the analysis of Arg and Lys methylation sites.

The work from Maron and colleagues (Doi:10.1016/j.isci.2021.102971) reports a thorough analysis of Arg-methylated proteins and shows the complex interplay between type I and type II PRMTs. Interestingly, the authors implemented a decision-tree-based MS method to optimize the identification of Arg-methylated peptides. The presence of internal basic residues in Arg methylated peptides sequesters protons, thereby

reducing their mobility along the peptide backbone and preventing dissociation using CID/HCD. The potential of ETD for improving the fragmentation of such highly charged peptides was first demonstrated using synthetic methylated peptides (PMID: 19110445) and on PIWI proteins (PMID: 19584108). Capitalizing on the capabilities of a modern Tribrid instrument, Maron and colleagues unleashed the potential of such strategy and achieved efficient peptide backbone fragmentation for highly charged (7+) and heavily methylated peptides (9xRme). The instrumentation available in our lab only permitted HCD fragmentation leading to a potential bias in our data sets towards shorter and less modified methylated-peptides. To compensate for such limitation, we pre-fractionated our peptide samples using reverse phase at high pH prior to immunoprecipitation. As a result, we were able to identify, to our knowledge, one of the largest catalogues of Arg methylation reported to date, comprising 5,931 R-methylations (Rme1, Rme2, ADMA, SDMA) in 1,558 proteins (2,443 R-methylations in 585 proteins were reported by Maron). The sensitivity of our analysis improved the detection of less abundant non-RG motifs (Suppl. Figure 5D in our original submission). The results of Maron and colleagues are in great agreement with our data regarding the sequence preference of Arg methylation, including Pro-directed sequences Prmt4 (Carm1) (Suppl. Table 6 in our original submission) and the enrichment in intrinsically disordered regions of proteins (Suppl. Figure 5C in our original submission). We have now cited this work on Page 3, line 74 and line 86 of the revised manuscript file.

On the other hand, the studies by Fong et al. (doi:10.1016/j.ccell.2019.07.003) and Fedoriw et al (doi:10.1016/j.ccell.2019.05.014) in addition to Gao et al (using non-MS-based approaches in this case) (doi: 10.1093/nar/gkz200) showed that combined Prmt5 (Type II) and Type I Prmt inhibition results in synergistic anti-tumoral effects. Mechanistically, both studies showed that Type I and Type II Prmts regulate numerous splicing proteins, and therefore, inhibiting their enzymatic activities reduces splicing fidelity and results in preferential killing of splicing-mutant leukemias (Fong et al) and MTAP-deficient cancer cells (Fedoriw et al.). Therefore, their results are in agreement with a major finding from our study concerning the changes in thermal stability of splicing proteins found in mESCs upon Prtm5 inhibition (Page 6, Lines 190-192 in the original submission file). To gain further insight, we have compared our thermal stability data with the immuno-precipitation-based results from Fong and Fedoriw. Despite obvious differences between mESCs and the cancer lines analyzed in these two studies, out of the 22 and 78 regulated proteins identified by Fong and Fedoriw, 8 and 19 proteins also had altered stability in our data, respectively. Most of them are RNA binding proteins (FDR 2.5E-15) and included several splicing factors such as Snrpb, Syncrip, Alyref, Taf15, Fus and Wdr33. Remarkably, our work revealed other splicing proteins with altered stability, indicating the great complementarity of both approaches. We have now cited both studies on Page 6, line 172 and Page 12, line 419 in the revised manuscript file.

Lastly, the recent work of Li et al. focused on Prmt7, the only Type III Prmt, and reports on the identification of the Prmt7-regulated methylome. Interestingly, the authors also found that Prmt7 regulates numerous RNA binding proteins including proteins of the spliceosome rich in RG motifs. We have compared our results with those from Li et al and found that 10 out of the 85 Prmt7-regulated proteins had altered thermal stability in our Prmt5 data. Not surprisingly, these 10 proteins were also enriched in RNA binding and splicing. When comparing all these studies, we noted that some splicing proteins were found regulated by Prmt1, Prmt5 and Prmt7 suggesting that these enzymes form an intricate regulatory network controlling RNA processing-related processes, acting in some cases redundantly on the same proteins. We have cited this work on Page 12, line 419 in the revised manuscript file.

Minor critique:

- Figure 1A is not arranged to easily discern ESCs vs MEFs. I suggest considering another arrangement, maybe two separate plots, rather than relying on the subtle color differences.

Figure 1A represents the relative differences between MEFs and mESCs, as well as an estimation of the abundance levels for all methyl-transferases and de-methylases identified in our data using iBAQ (intensity-based absolute quantification). However, we acknowledge the reviewer's feedback that the current display is suboptimal because the protein levels for mESCs and MEFs are not clearly depicted. While creating two separate plots may make direct protein comparison difficult, we have made efforts to improve the figure's interpretation. We have re-arranged all the elements and simplified the color scheme to better distinguish mESCs and MEFs data, resulting in enhanced readability. We appreciate the reviewer's suggestion, which has helped to clarify our findings.

- Similarly, Fig 1C would be clearer if arranged vertically so reading the MTase names would be simpler (like in all the subsequent figures with vertical heatmaps).

We have restructured Figure 1 to improve its presentation. Initially, we arranged Figure 1C horizontally to accommodate all other panels. However, given that Figure 1C contains the PRM results that validate our findings from the global proteome analysis (Figure 1A), we have now moved Figure 1C to the Supplementary Material. Figure 1C is Suppl. Figure 1B in the revised version and is arranged vertically.

- Line 138 has a citation for the association with Wdr77 (#27) that is about *C. elegans* PRMT5 which does not have an associated cofactor. Change the citation to another report or a review.

This was a mistake and we thank the reviewer for bringing to our attention an error in our reference. We have now replaced this reference for PMID:11756452 (Page 4, line 121 in the revised manuscript)

- On line 156, it is noted that Alyref and Snrpnb interact with PRMT5. Citations are necessary for these specific claims.

Both Prmt5 interactors were retrieved from the Biogrid database and are derived from this publication PMID: 33961781. We have now placed this reference on Page 5, line 139 in the revised manuscript file.

- On line 170, it is not clear what "...R468me is highly stoichiometric" means. This use of the word is also found in line 383. This is not a typical use of the word. Please define – does it mean that the residue is highly modified? Or what?

We apologize if our terminology was unclear. Stoichiometry refers here to the fractional occupancy of a particular methylation site. Indeed, a highly stoichiometric methylation site means that is highly modified. The term stoichiometry in the context of post-translational modification analysis has been normally used in the several proteomics publications (see for instance PMIDs: 20068231 and 30837475 for the cases of phosphorylation and acetylation respectively). We have now clarified this in the text (Page 5, line 153-154).

- On line 195, what does "we did not see such trend for non-polyadenylated RNA..." mean? I suggest this be defined more clearly, as it could be an important point. But what are the authors addressing? rRNAs? snoRNAs? snRNAs? tRNAs? Histone genes? Define it more clearly.

We are sorry for the confusion with this statement. A large fraction of the studies reporting the identification of protein-RNA interactions are focused towards poly-adenylated RNAs (mRNAs) (PMID: 29339797). However, a recent report from the Krijgsveld Group (PMID: 30528433) introduced a novel approach termed XRNAX that is able to purify all RNA biotypes including tRNA, rRNA, small nuclear RNAs (snRNAs), and small nucleolar RNAs (snoRNAs). This work significantly expanded the repertoire of RNA binding proteins and classified them as those that bind poly-adenylated RNAs and non-poly-adenylated RNAs (rRNA, snoRNA, tRNA...).

In our study, we investigated the changes in thermal stability of RNA-binding proteins upon Prmt5 inhibition, utilizing the classification made by Krijgsveld and colleagues. Our results revealed a significant alteration in the poly-adenylated RNAbp. We have provided further clarification regarding this matter in the revised manuscript file on Page 6, line 178-180.

Reviewer #2 (Remarks to the Author):

The study by Sayago et al represents an impressive effort in mapping the effect of protein methylation in mouse embryonic stem cells and comparing that to mouse embryonic fibroblasts. It is shown convincingly that inhibition of demethylase activity changes the thermal stability of several hundred proteins thus likely modulating their function. Follow up *in vitro* assays further cement these findings by identifying a novel PRMT5 substrate. The complementarity of the method to standard abundance proteomics is also shown by lack of correlation between changes of the two measurements. The data reveals multiple proteins related to stress granule formation and it is shown that arsenite treatment in the presence of the inhibitor modulates the extent of granule formation. Further work with EZH2 inhibitors reveals links to mitotic proteins and reveals a role of EZH2 in mitotic chromosomes. This is impressive work and will be widely used in the proteins methylation field I believe. I recommend publication following minor changes.

We would like to express our gratitude to the reviewer for providing positive feedback on our study, which we believe has further strengthened the manuscript. We are delighted to hear that the reviewer found our efforts in mapping the effect of protein methylation to be impressive. We appreciate his/her recommendation for publication following minor changes, which we have addressed below.

Comments:

Figure 5F,H,G plots: are these multiple replicates? If yes, it would be better to analyze with NPARC approach to deduce p-values for differences, rather than summing up.

We believe that the reviewer may have been referring to Figure 3F, H, G (rather than Figure 5F, H, G). Figure 3F,H,G shows the *in vitro* thermal stability assay of methylated vs non-methylated G3bp3 (and Prmt5 and Wdr77) of a single experiment. The experiment was repeated independently, and the findings were confirmed (Suppl. Fig 8 in the original submission). These results demonstrated that the engagement of GSK595 led to a clear stabilization of Prmt5-Wdr77 at 53°C and 55°C. In contrast, G3bp2 did not show any significant changes, as discussed on Page 6, lines 180-188 of the original submission. To provide a more comprehensive representation of the data, the figure included the aggregated remaining soluble protein at each temperature, similar to the PISA approach.

We concur with the reviewer on the importance of calculating the actual p-value of the difference between these melting curves. However, we would like to clarify that due to the limited amount of protein substrate available, we were only able to assess five temperatures, two of which were at the extremes (37°C, representing the fully soluble state, and 78°C, representing the fully insoluble state). This could potentially impact the melting curves fitting. Nonetheless, we utilized the non-parametric analysis of response curves (NPARC) to analyze these data (see below) and found a significant difference in the melting curves of Prmt5 and Wdr77 upon addition of GSK595. On the other hand, no significant difference was observed for G3bp2. In the revised figure, we have merged the results from both replicates (Suppl. Fig 8 has been consequently removed) and included the computed p-values in the new Figure 3F, 3H, and 3G. Text has been adjusted accordingly (Page 5, lines 164-168 in the revised manuscript). We believe the presentation of the data has improved and we thank the reviewer for the suggestion.

“GSK595 alone however did not promote the formation of stress granules but led to alterations in nucleolar fragmentation” could you discuss whether any nucleolar proteins are specifically affected in stability?

Among the proteins showing altered stability upon Prmt5 inhibition (changes common to both GSK591 and GSK595), there was indeed a significant enrichment in nucleolar-related processes such as Ribosome biogenesis (GO:0042254) (FDR=2.7E-33) (Suppl. Table 6). Specifically, there were 124 proteins that are annotated by GO-Cellular Component as “Nucleolus” (GO:0005730). However, it is important to note that a fraction of these nucleolar proteins have been reported as insoluble in previous studies, as discussed in the next comment to this reviewer. Among the nucleolar soluble proteins with altered stability in our data there were several subunits of the RNA pol I involved rRNA transcription, the fibrillar protein FBLL1 involved in rRNA methylation and also Arg-methylated proteins such as EDF1 which coordinates the cellular responses to ribosomal collisions upon translation of aberrant mRNAs (PMID: 32744497) and Wdr33, the key subunit of the cleavage and polyadenylation specificity factor (CPSF) complex in the recognition of the polyA signal (PMID: 25301781). Therefore, the changes in stability and, likely, solubility measured in nucleolar proteins are consistent with the nucleolar fragmentation observed by immuno-fluorescence and underscore the role of Prmt5 and Arg methylation in the regulation of this membrane-less organelle. The authors would like to thank the reviewer for bringing attention to this point. We have updated the text to reflect this discussion (Page 7, line 220 in the revised manuscript file).

For discussion section: PISA assay does not differentiate between stability and solubility changes, so it is conceivable that some proteins linked to LLPS are changing more in solubility than stability (see eg here: PMID: 35864335, PMID: 30858367), this would be good to mention in the discussion section.

We are grateful to the reviewer for drawing our attention to the potential impact of solubility on the observed variations in thermal stability. It is possible that certain alterations in stability in our study may be attributed to variances in solubility, which may lead to varying degrees of efficiency in protein extraction using the NP-40 buffer utilized in our PISA experiments.

To get further insight into this matter, we have compared our data with PMID: 35864335. Sridharan et al implemented an elegant approach to identify proteins associated with biomolecular condensates and examined how their solubilities were affected upon RNase treatment. Among the 8,104 proteins assessed by PISA in our Prmt5 study, we were able to retrieve solubility profiles for 4,322 of them. These 4,322 proteins showed no biases in the proportion of soluble (82%), insoluble RNA-sensitive (5%) and insoluble RNA-insensitive (15%) proteins (Figure A) as determined by Sridharan. However, the partition of soluble and insoluble proteins was markedly different between PISA-stabilized and destabilized proteins (Figure A). We found that 43% of the PISA-stabilized proteins have been reported as insoluble proteins by Sridharan, a 12-fold and 2-fold increase compared to PISA-destabilized (3.5%) and all measured (18%) proteins. Nonetheless, we only observed a subtle difference in the proportion of RNA-sensitive and RNA-insensitive insoluble proteins between PISA-stabilized (37%) and all measured proteins (30%).

Among the insoluble-RNA sensitive proteins that we found with increased stability in our data, there were FBL, NPM1 and NOP56. However, some of the proteins that Sridharan found to undergo complete solubilization upon RNase, such as HNRNPA1 and PRPF6, did not show apparent stability changes in our data. We also performed functional analyses of PISA stabilized proteins and found that both soluble and insoluble proteins were enriched in RNA binding proteins (Figure B). Despite the differences between our study and the work from the Savitski lab, such as variations in cell lines and experimental set-up, our comparisons suggest that a subset of the PISA-stabilized proteins in our data may, in fact, exhibit an increase in solubility. Nevertheless, these comparisons also indicate that the stability changes we observed in RNA binding proteins upon Prmt5 inhibition cannot be solely attributed to increased protein solubility.

Finally, the work of Sridharan also explores the role of phosphorylation in the regulation of molecular condensates. Given the known interplay between Arg methylation and phosphorylation as regulators of phase separation and RNP granule dynamics (PMID: 30587571), we believe that future analyses of stability, solubility and abundance changes accounting for cross-talk between these two protein modifications (PMID: 33856219) will shed some light into the mechanisms mediating in their assembly/disassembly.

In the revised version, we have discussed this possibility (Page 13, lines 426-429) and included the figure shown below as new Suppl Figure 22.

A

B

PISA-stabilized (Sayago et al.) & Insoluble proteins (Sridharan et al.) PISA-stabilized (Sayago et al.) & Soluble proteins (Sridharan et al.)

Reviewer #3 (Remarks to the Author):

The manuscript titled "DECODING PROTEIN METHYLATION FUNCTION WITH THERMAL STABILITY ANALYSIS" elaborated versatile features of protein methylation and identified several substrates of Prmt5 and Ezh2 in mESCs and MEFs mainly through analyzing the alterations in thermal stability of proteins. Although the approach of decoding protein methylation function proposed in this article has a promising application prospect, the proteins analyzed here are still not comprehensive and lack sufficient biochemical experimental data. In addition, multiple typos are found in the current version. Some specific concerns are outlined below:

We are pleased to hear that the reviewer considers our approach to be promising in terms of its application prospects. We apologize for any typos that may have been present in the manuscript. We have thoroughly revised the text and ensured that all errors had been corrected. While we appreciate the reviewer's critique regarding the significance in gaining mechanistic insights into the molecular consequences of protein methylation for some of the identified targets, the authors feel that pursuing such a goal may deviate from the primary message we intended to convey. Here, we aimed to provide a framework based on thermal stability for the large-scale analysis of protein methylation. To this end, we conducted a thorough proteomic exploration of the potential of this approach to study both Arg and Lys methylation, which are known to regulate distinct biological processes. We targeted two methyl-transferases Prmt5 and Ezh2 that have important roles in pluripotent mESCs and compared these results with those of differentiated MEFs. Our approach involved (i) profiling global thermal stability changes to gain an overview of their potential downstream effects, (ii) identifying novel methylated substrates, such as G3bp2 and Mki67, and (iii) demonstrating their involvement in regulating specific biological processes, such as stress granules and mitotic chromosome condensation. Therefore, our data show that thermal stability analysis serves as a proxy to uncover functions controlled by protein methylation, which is a valuable complement to immunopurification-based approaches. In addition, our study also reports the identification of over 6,000 methylations in mESCs and MEFs (Suppl. Table 6). To our knowledge, more than half of these methylations are novel. In conclusion, our study significantly expands the scope of proteins and biological functions that might be under the control of this modification and represents a rich resource for those interested in the methylation field, facilitating the discovery of potential methyl-transferase substrates.

1. This manuscript identified G3bp2 as a novel Prmt5 substrate, however, an article titled "USP7- and PRMT5-dependent G3BP2 stabilization drives *de novo* lipogenesis and tumorigenesis of HNSC" has recently been published on the journal *Cell death & disease* and this article elaborated the function of Prmt5-mediated G3bp2-R468me2. Suggest to carefully read this article and make appropriate adjustments of the relevant part in your manuscript.

We apologize for the oversight of not citing the work by Wang and colleagues, which, to our knowledge, it was published online after submitting our manuscript. We thank this reviewer for bringing this article to our attention. In this study, Wang and colleagues report that Prmt5 di-methylates G3bp2 in R468 in head and neck squamous carcinoma (HNSC) cell lines. This is in great agreement with our *in vitro* MTase assay, as we found R438me1, R468me and R468me2 in G3bp2, with the latter being over ten times more abundant than the other two sites (see figure below). In addition, Wang *et al.* also showed that G3bp3-R468me2 promotes its de-ubiquitination by USP7 in HNSC. However, according to thebiogrid.org, Usp7 has not been identified as an interactor of G3bp2 in other cell types. Additionally, in a systematic mapping of deubiquitinating enzymes co-authored by us, G3bp2 was not listed among the substrates of Usp7 (PMID: 34063716) (<https://ehubio.ehu.eus/dubase/>). Furthermore, Wang and colleagues showed that R468me2-Prmt5 in HNSC activates *de novo* lipogenesis. However, our analysis of proteome abundance data did not reveal any GO term related to lipid metabolism among the proteins that decreased abundance upon Prmt5 inhibition in mESCs and MEFs. Moreover, ACLY, FASN, ACSL3 and other enzymes remained unaffected in our cell types. Thus, based on these observations, whether a similar mechanism to the one reported by Wang for G3bp2-R468me2 in HNSC is also present in mESCs will need to be experimentally assessed. As suggested by the reviewer, we have now cited this study and modified the text accordingly (Page 5, line 150 and Page 12, lines 386-391 in the revised manuscript file). We have included the figure shown below as Supplementary Figure 8A in the revised Supplementary Material. We have also updated the section header "G3bp2 is a novel Prmt5 substrate..." to "G3bp2 is Prmt5 substrate..." (Page 5, line 144 in the revised version) and corrected Figure 5 legend (page 27, line 923).

Figure legend. The intensity of mono- and di-methylation identified in R438 and R468 of G3bp2 in the *in vitro* methyl-transferase assay of Prmt5 is presented. A.U. arbitrary units.

2. Suggest to provide a reasonable explanation of the data about mono-methylation in R468 of G3bp2 in figure 3B because the level of R468me is significantly elevated in the presence of GSK595.

This is an interesting question, and we thank the reviewer for pointing this out. Prmt5 is a methyl-transferase that catalyzes di-methylation in a distributive fashion (PMID: 33274632). Briefly, Prmt5 monomethylates a protein substrate and is released into solution before rebinding to carry out the second methylation to form the SDMA product. The GSK595 inhibitor used in our *in vitro* methyl-transferase assay shown in Figure 3 is a protein substrate competitive inhibitor (SAM uncompetitive inhibitor). Therefore, our results are in agreement with this kinetic model and indicate that the GSK595 has an inhibitory effect over the binding of Prmt5 on the mono-methylated substrate (see figure below). This explains the increase of R468me1 and the decrease of R468me2 in the presence of GSK595. Remarkably, the decrease observed in the unmodified counterpart peptide in the Prmt5 condition (Figure 3C) indicates that the stoichiometry of R468me2 (i.e. fractional occupancy of this modification) is significantly higher than the stoichiometry of R468me1 found in the Prmt5+GSK595 condition. We have clarified these results on Page 5, lines 154-156 in the revised version of the manuscript. We have included the schematic model for Prmt5 methylation as Suppl. Figure 8B in the revised Suppl. Material.

Figure legend. Top panel, Prmt5 methylates protein substrates in a distributive manner. Bottom panel, GSK595, a substrate competitive Prmt5 inhibitor, prevents its binding to the mono-methylated substrate and thereby the subsequent addition of a second methyl group.

3. Although the manuscript gave the reasons why the thermal stability of G3bp2 only has small difference, the author should summarize the advantages and disadvantages of this approach and give some perspectives on the improvement of this approach.

The assay shown in Figure 3D-H was designed to validate the results of our large-scale thermal stability analysis. Although our PISA results indicated that inhibition of Prmt5 significantly stabilized G3bp2, and we identified G3bp2-SDMA in mESCs and *in vitro* by Prmt5, we could not eliminate the possible contribution of other factors, such as other post-translational modifications (PTMs).

Therefore, the main advantage of this assay is that it allows us to measure the actual impact of methylation on the thermal stability of a protein in an isolated system with only the methylation pattern as the variable, without other confounding factors. However, it is worth noting that the *in vitro* conditions used in the assay

differ from the physiological conditions of the protein inside the cell, where it may require the coordinated action of other interacting proteins to function, and even RNA in the case of G3bp2.

An alternative approach to our assay could be the co-transfection of both the protein substrate and methyltransferase (including both active and mutant inactive versions), followed by the immunoprecipitation of the complexes and assessment of the melting curves for each case. This would better approximate the *in vivo* environment and help determine the impact of methylation on the protein thermal stability.

We addressed these questions in the revised version of the manuscript on Page 12, lines 393-401.

4. Many protein names in the article appear in the form of abbreviations, which creates obstacles to the reading of the manuscript, it is recommended to give the full name and the corresponding abbreviation when the protein first appears.

We are sorry for this issue. We have now included the full protein name, along with its official gene symbol, the first time that a protein is mentioned in the text.

5. There are many "'s" in the manuscript and suggest to change them to "s". Line 284 and 289, change "DNMT's" to "DNMTs"; Line 304, change "Ezh2i's" to "Ezh2is".

We have addressed this matter throughout the manuscript.

6. In the section of "A non-canonical function of Ezh2 in chromosome organization", the author should first conclude the canonical function of Ezh2 and then show the non-canonical function of Ezh2 in chromosome organization to make this part more clearly.

Ezh2 is the catalytic subunit of the Polycomb Repressive Complex 2 (PRC2), responsible for depositing H3K27me3, an important repressive epigenetic mark. As expected, inhibition of Ezh2 with both GSK126 and EPZ-6438 led to a marked decrease in H3K27me3 in mESCs and MEFs. These results were introduced on Page 8, lines 246-247 and Suppl. Figure 12 in the original submission. In response to the reviewer's feedback, we have now highlighted the Ezh2 canonical function at the beginning of the section "A non-canonical function of Ezh2 in chromosome organization" on page 9, lines 290-291 in the revised manuscript.

7. Suggest to define a relationship between the changes of thermal stability and the function of the protein methylation. When using inhibitors of Prmt5 and Ezh2, some proteins with slight alterations of thermal stability compared to their methylated forms are defined as substrates and exhibit critical functions. However, it is unreasonable to conclude that proteins with changes of thermal stability regardless of the extent of the changes are all substrates or have significant functions when they are methylated.

Our study revealed hundreds of proteins with altered thermal stability upon inhibition of the enzymatic activity of Prmt5 and Ezh2. In the case of Prmt5, a significant fraction of these proteins was found in our analyses to be methylated in Arg, including known Prmt5 substrates (Figure 2F). It is worth noting that our objective was not to assert that all proteins with altered stability are necessarily downstream substrates. Instead, our aim was to emphasize that thermal stability analysis is a useful method for identifying potential substrate candidates (see, for instance, Discussion section, Page 11, lines 376-378 in the original submission). Certainly, the identification of protein substrates would require additional experimental data. In our work, we followed up two of these proteins and demonstrated that Prmt5 and Ezh2 methylate G3bp2 and Mki67 in specific motifs, consistent with known substrate recognition preferences of each enzyme. Furthermore, in the case of Ezh2-Mki67, we showed that they physically interact in mESCs. Collectively, our findings suggest that G3bp2 and Mki67 may be novel substrates of Prmt5 and Ezh2, respectively. Similarly, we hope that our data sets will be valuable for other researchers seeking to identify new substrates of Prmt5 and Ezh2 as well as for other methyl-transferases using the strategy described here. We have modified the text to make this point clearer (Page 14, lines 467-469). We have also changed the section header "Arg methylated proteins, including known-substrates, show alterations in thermal stability upon inhibition of Prmt5 in mESCs." to "Prmt5 inhibition in mESCs leads to thermal stability changes of numerous Arg-methylated proteins, including known-substrates." which more accurately reflects our findings (Page 4, lines 97-98).

8. The proteins can be methylated presented in the manuscript are all increased in thermal stability values when using inhibitors of Prmt5 and Ezh2, however, this is not comprehensive enough. The changes of

thermal stability include both increases and decreases, so suggest to analyze some proteins with decreased thermal stability values and clarify the relationship of such changes and function.

Among the proteins that decreased stability, we found several Prmt5 interactors, including known substrates (Figure 2F). Similarly, we also showed that certain epigenetic regulators of specific histone marks were destabilized upon Ezh2 inhibition (Figure 5B). In addition, functional enrichment analyses for stabilized and destabilized proteins can be found in Figure 4E and Suppl. Table 8 for Prmt5 and in Suppl. Figure 15A and Suppl. Table 9 for Ezh2 in the original submission files (note that Suppl. Figure 15A is 14A in the revised Suppl. Material).

In the case of Prmt5, we found a clear enrichment in RNA-related processes among the stabilized proteins (see figure below, panel A). Destabilized proteins showed enrichment in terms related to mitochondria and metabolism (panel B). However, we noticed that many of these mitochondrial proteins were driven by differences in protein abundance rather than differences in stability (Suppl. Table 7). Regarding Ezh2, we observed a clear enrichment in cell cycle and chromosome organization among the stabilized proteins (panel C). On the other hand, destabilized proteins showed enrichment in nucleotide metabolism and other general metabolism-related terms (panel D). The connection between Ezh2 and nucleotide metabolism observed in our data might be in agreement with a pre-print study showing that cytosolic Ezh2 interacts with the Inosine-5'-monophosphate dehydrogenase 2 (Impdh2) in the cytosol of melanoma cells to regulate GTP production (doi: <https://doi.org/10.1101/2021.11.02.467024>). Remarkably, we identified Impdh2 as one of the most significant Ezh2 interactors in mESCs. Furthermore, we observed that the interaction between Impdh2 and Ezh2 was significantly enhanced in the presence of Gsk126, as shown in Supplementary Table 10.

Therefore, these results revealed a higher level of functional co-regulation among the stabilized proteins, as opposed to those that experienced decreased stability. As a result, we focused our work on investigating the stabilized proteins in greater detail. We appreciate the feedback provided by the reviewer and recognize the significance of presenting the results for destabilized proteins as well. To provide a more comprehensive understanding of the data, we have highlighted some of the key findings related to destabilized proteins in the revised manuscript (Page 6, lines 172-176 and Page 8, lines 249-251).

Figure legend. Stabilized and destabilized proteins upon inhibition of Prmt5 and Ezh2 were analyzed with StringDB. Functional links (high confidence) between proteins lists and the average node degree are shown. The GO-biological processes (sorted by FDR) significantly enriched in each case are displayed.

The items below are several mistakes which need revising:

- 1.Line 31: change "others" to "other".
- 2.Line 237: change "SMDA" to "SDMA".
- 3.Line 290: change "its" to "their".
- 4.Line 337-338: delete "in" because this word appears twice in a row.
- 5.Line 345: change "possess" to "possessed" or "possesses".
- 6.Line 404: change "involve" to "involves".
- 7.Line 419: change "Ezh2 is a KMTs" to "Ezh2 is a KMT".
- 8.Line 434: change "abundances" to "abundance" because abundance belongs to uncountable nouns.
- 9.Line 446: delete "that" because this word appears twice in a row.
- 10.Line 752: change "2%SDS" to "2% SDS".
- 11.Line 762: change "4C" to "4°C".

All these mistakes have been now corrected.

It is recommended to go over the manuscript carefully again to avoid grammatical errors. Moreover, some sentences do not conform to the characteristics of English writing. For example, the sentence in line 436-437. Suggest to change this sentence to "Our results are then in great agreement with a recent report showing that PRC2 is required for chromosome compaction during mitosis in mESCs". In summary, the language of the manuscript needs to be polished.

We apologize for the grammatical errors present in our manuscript. In the revised version, we have thoroughly checked for and corrected all of these errors, including the specific sentence mentioned by the reviewer.

REVIEWERS' COMMENTS

Reviewer #1 (Remarks to the Author):

The authors have satisfactorily addressed my critique.

Reviewer #2 (Remarks to the Author):

I am very happy with the revision and suggest publication.

Reviewer #3 (Remarks to the Author):

I have no other concerns.